# Overcoming Knowledge Barriers: Online Imitation Learning from Visual Observation with Pretrained World Models

**Xingyuan Zhang[1, 2]\***, **Philip Becker-Ehmck[1]**, **Patrick van der Smagt[1,3]**, **Maximilian Karl[1]**
**[1]Volkswagen Group, [2]Technical University of Munich, [3]Eötvös Loránd University Budapest**
`xingyuan.zhang@tum.de, philip.becker-ehmck@volkswagen.de`

Reviewed on OpenReview: `https://openreview.net/forum?id=BaRD2Nfj41`

## Abstract

Pretraining and finetuning models has become increasingly popular in decision-making. But there are still serious impediments in Imitation Learning from Observation (ILfO) with pretrained models. This study identifies two primary obstacles: the Embodiment Knowledge Barrier (EKB) and the Demonstration Knowledge Barrier (DKB). The EKB emerges due to the pretrained models' limitations in handling novel observations, which leads to inaccurate action inference. Conversely, the DKB stems from the reliance on limited demonstration datasets, restricting the model's adaptability across diverse scenarios. We propose separate solutions to overcome each barrier and apply them to Action Inference by Maximising Evidence (AIME), a state-of-the-art algorithm. This new algorithm, AIME-NoB, integrates online interactions and a data-driven regulariser to mitigate the EKB. Additionally, it uses a surrogate reward function to broaden the policy's supported states, addressing the DKB. Our experiments on vision-based control tasks from the DeepMind Control Suite and MetaWorld benchmarks show that AIME-NoB significantly improves sample efficiency and converged performance, presenting a robust framework for overcoming the challenges in ILfO with pretrained models. Code available at `https://github.com/IcarusWizard/AIME-NoB`.

## 1 Introduction

We have been going through a paradigm shift from learning from scratch to pretraining and finetuning, in particular in Computer Vision (CV) (He et al., 2016; Radford et al., 2021; He et al., 2022) and Natural Language Processing (NLP) (Devlin et al., 2019; Radford et al.; Ouyang et al., 2022; Touvron et al., 2023a;b) fields due to the increasing availability of foundation models (Bommasani et al., 2021) and ever-growing datasets. However, it is still unclear how to adapt this new paradigm into decision-making, in particular what type of models we need to pretrain and how these models can be adapted to solve downstream tasks. Recent work (Zhang et al., 2023; DeMoss et al., 2023; Sekar et al., 2020; Rajeswar et al., 2023; Hansen et al., 2023a) showed that pretrained latent space world models enable successful and efficient transfer to new tasks with either reinforcement learning (Sekar et al., 2020; Rajeswar et al., 2023; Hansen et al., 2023a) or Imitation Learning from Observation (ILfO) (Zhang et al., 2023; DeMoss et al., 2023). ILfO (Torabi et al., 2018; 2019; Baker et al., 2022; Zhang et al., 2023; DeMoss et al., 2023; Liu et al., 2022a), especially from videos (Baker et al., 2022; Zhang et al., 2023; Liu et al., 2022a; DeMoss et al., 2023), is a more promising approach in this new paradigm since it does not require a handcrafted reward function which is hard to define for many real-world tasks.

But there are challenges when using pretrained models in ILfO. To quantify these, we introduce two new barriers, which we call the Embodiment Knowledge Barrier (EKB) and the Demonstration Knowledge Barrier (DKB). The EKB describes the limitation of a pretrained model when confronted with novel observations and actions beyond its training experience. The DKB describes the generalisation from a limited number of

---

\*Corresponding author.

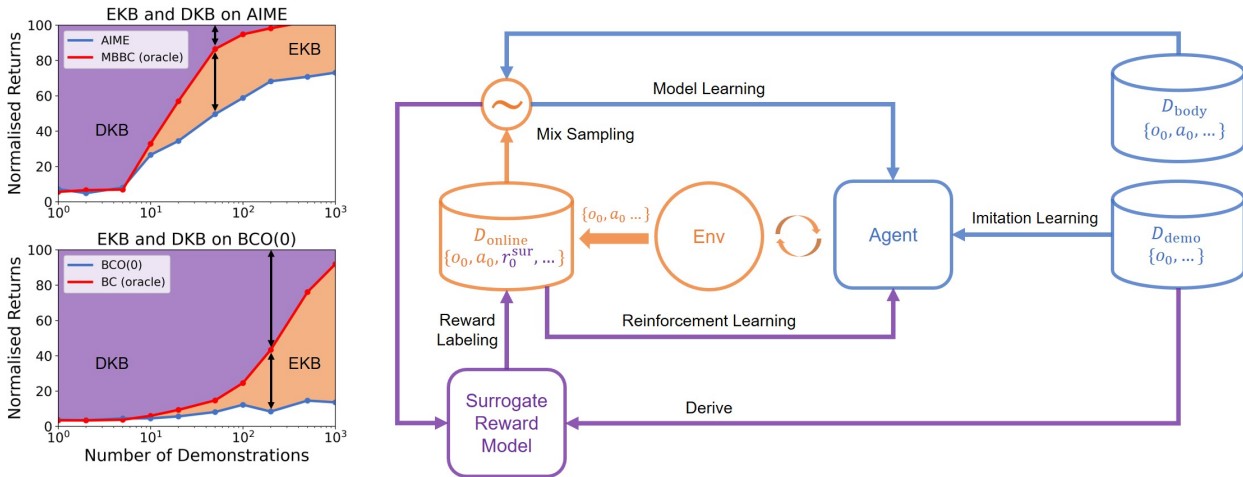

Figure 1: Main idea of this paper. On the left, we plot the performance of BCO(0) and AIME together with their oracle versions, which remove the EKB, w.r.t. different number of demonstrations on walker-run task. For each setting with the same number of the demonstrations, i.e. each column, the value difference between the oracle version and the expert is the Demonstration Knowledge Barrier (DKB) while the value difference between the algorithm and its oracle version represents the Embodiment Knowledge Barrier (EKB). On the right, we present the solutions proposed in this paper to overcome the two barriers. The blue parts represent the original version of the algorithms that suffer from the knowledge barriers. Orange parts demonstrate the solution for EKB, where the agent is allowed to interact with the environment and use $D_{\text{online}}$ together with $D_{\text{body}}$ to update the world model. Purple parts show the solution for DKB, where a surrogate reward model is derived from $D_{\text{demo}}$ and used to label $D_{\text{online}}$ and then used as an RL signal for policy learning.

expert demonstrations in imitation learning (Ho & Ermon, 2016). State-of-the-art approaches such as BCO(0) (Torabi et al., 2018) and AIME (Zhang et al., 2023) typically suffer from these two knowledge barriers. First, these algorithms depend on the pretrained model to infer missing actions from observation sequences. Thus, when the model has not seen a specific observation before, it may not know enough about the embodiment to infer the correct action. Second, if the policy optimisation is only guided by limited demonstrations, it can lead to a policy that generalises poorly, working well in some states but not in others.

To better showcase the two barriers, in Figure 1, we evaluate both AIME (Zhang et al., 2023) and BCO(0) (Torabi et al., 2018) and their oracle versions w.r.t. different number of demonstrations on walker-run task. Both algorithms pretrain a model from a large embodiment dataset and use that to infer the actions for the observation-only demonstrations. The oracle versions remove the need to infer the missing actions by providing the algorithm with the true actions, thus removing the EKB. As we can see from the figure, for each setting with the same number of demonstrations, the two algorithms are always upper-bounded by the corresponding oracle version, and the difference between them represents the EKB. On the other hand, even if given the true actions of the expert, imitation performance may still be impacted by a limited number of demonstrations providing insufficient coverage of the state space. Thus, the difference between the oracle version and the expert performance represents the DKB.

In this paper, we study how to overcome these barriers to improve the performance of ILfO approaches with pretrained models, in particular of AIME. For the EKB, we extend the setting from offline to online by allowing the agent to further interact with the environment to gather more data to train the world model. While for the DKB, we introduce a surrogate reward function to allow the policy to essentially train on more data. We demonstrate that the proposed modifications effectively overcome the two barriers and significantly improve the performance on nine tasks in DeepMind Control Suite (DMC) (Tunyasuvunakool et al., 2020) and six tasks in MetaWorld (Yu et al., 2021).

We summarise our contributions as follows:

- We identify and thoroughly analyse the two knowledge barriers, namely EKB and DKB, in the current pretrained-model-based ILfO methods.

- We propose AIME-NoB as an extension of the state-of-the art AIME algorithm by resolving the two knowledge barriers. Specifically, AIME-NoB uses online interaction with a data-driven regulariser to overcome the EKB and learn a surrogate reward function enlarging state coverage to overcome the DKB.

- We evaluate AIME-NoB on 15 tasks from two vision-based benchmarks and the results demonstrate AIME-NoB significantly outperforms previous state-of-the-art methods both in terms of final performance and sample efficiency. We also conduct thorough ablation studies to show how the EKB and the DKB are overcome by the proposed modifications and how different design choices influence the performance.

## 2 Preliminary

We mostly follow the problem setup as described in Zhang et al. (2023). We consider a POMDP problem defined by the tuple $\{S, A, T, R, O, \Omega\}$, where $S$ is the state space, $A$ is the action space, $T : S \times A \rightarrow S$ is the dynamic function, $R : S \rightarrow \mathbb{R}$ is the reward function, $O$ is the observation space, and $\Omega : S \rightarrow O$ is the emission function. The goal is to find a policy $\pi : S \rightarrow A$ which maximises the expected accumulated reward, or return, i.e. $\mathcal{R}(\pi) = \mathbb{E}_{a \sim \pi}[\sum_{t=1}^{T} r_t]$. Since in this work we focus on imitation learning, this oracle reward is not available to the agent. We mainly use this reward to quantify the performance of the learnt policies.

We presume the existence of three datasets of the same embodiment available to our agent. The *embodiment dataset* $D_{\text{body}}$ contains trajectories $\{o_0, a_0, o_1, a_1 \dots\}$ that represent past experiences of interacting with the environment. This dataset provides information about the embodiment for the algorithm to learn a world model. In addition, we also allow the agent to interact with the environment to collect new data in a *replay buffer* $D_{\text{online}}$. Note that, although the simulator will give us the reward information, the agent is not allowed to use them, and we only use the reward for evaluation purposes. The *demonstration dataset* $D_{\text{demo}}$ contains a few expert trajectories $\{o_0, o_1, o_2 \dots\}$ of the embodiment solving a certain task defined by $R_{\text{demo}}$. The crucial difference between this dataset and the other two datasets is that the actions are not provided anymore since they are not observable from a third-person perspective. The goal of our agent is to learn a policy $\pi$ from $D_{\text{demo}}$ which can solve the task defined by $R_{\text{demo}}$ as well as the expert $\pi_{\text{demo}}$ who generated $D_{\text{demo}}$.

### 2.1 World Models

A World Model (Ha & Schmidhuber, 2018) is a generative model which models a probability distribution over sequences of observations, i.e. $p(o_{1:T})$. The model can be either unconditioned or conditioned on other factors, such as previous observations or actions. When the actions taken are known, they can be considered as the condition, i.e. $p(o_{1:T}|a_{0:T-1})$, and the model is called embodied (Zhang et al., 2023). In this paper, we consider variational latent world models where the observation is governed by a Markovian hidden state. This type of model is also referred to as a State-Space Model (SSM) (Karl et al., 2017; Hafner et al., 2019b;a; Becker-Ehmck et al., 2019; Klushyn et al., 2021). Such a variational latent world model involves four components, namely

$$\text{encoder } z_t = f_\phi(o_t), \qquad\qquad \text{posterior } s_t \sim q_\phi(s_t|s_{t-1}, a_{t-1}, z_t),$$
$$\text{decoder } o_t \sim p_\theta(o_t|s_t), \qquad\qquad \text{prior } s_t \sim p_\theta(s_t|s_{t-1}, a_{t-1}).$$

$f_\phi(o_t)$ is the encoder to extract the features from the observation; $q_\phi(s_t|s_{t-1}, a_{t-1}, z_t)$ and $p_\theta(s_t|s_{t-1}, a_{t-1})$ are the posterior and the prior of the latent state variable; while $p_\theta(o_t|s_t)$ is the decoder that decodes the observation distribution from the state. $\phi$ and $\theta$ represent the parameters of the inference model and the generative model respectively.

Typically, the model is trained by maximising the Evidence Lower Bound (ELBO) which is a lower bound of the log-likelihood, or evidence, of the observation sequence, i.e. $\log p_\theta(o_{1:T}|a_{0:T-1})$. Given a sequence of

observations, actions, and states, the objective function can be computed as

$$\text{ELBO} = \sum_{t=1}^{T} J_t^{\text{rec}} - J_t^{\text{KL}} = \sum_{t=1}^{T} \log p_\theta(o_t|s_t) - D_{\text{KL}}[q_\phi||p_\theta]. \tag{1}$$

The objective function is composed of two terms: the first term $J^{\text{rec}}$ is the likelihood of the observation under the inferred state, which is usually called the reconstruction loss; while the second term $J^{\text{KL}}$ is the KL divergence between the posterior and the prior distributions of the latent state. To compute the objective function, we use the re-parameterisation trick (Kingma & Welling, 2022; Rezende et al., 2014) to autoregressively sample the inferred states from the observation and action sequence.

In summary, a world model is trained by solving the optimisation problem as

$$\phi^*, \theta^* = \underset{\phi,\theta}{\text{argmax}}\, \mathbb{E}_{\{o,a\}\sim D_{\text{body}},s\sim q_\phi}[\text{ELBO}]. \tag{2}$$

## 2.2 Action Inference by Maximising Evidence (AIME)

AIME is a state-of-the-art algorithm that uses a pretrained world model to solve ILfO in an offline setting. Specifically, it uses the pretrained world model as an implicit inference model by solving for the best action sequence that makes the demonstration most likely under the trained world model. The imitation can be done jointly with the action inference using amortised inference and the re-parameterisation trick by solving the following optimisation problem

$$\psi^* = \underset{\psi}{\text{argmax}}\, \mathbb{E}_{o\sim D_{\text{demo}},s\sim q_{\phi^*,\theta^*},a\sim\pi_\psi}[\text{ELBO}], \tag{3}$$

where $\psi$ is the parameter for policy $\pi_\psi(a_t|s_t)$. The resulting objective is very similar to Equation (2), with a subtle difference of the sampling path. That is in the new objective, only the observations are sampled from the dataset and both states and actions are sampled iteratively from the learned model and the policy, respectively.

## 3 Methodology

In this section we will analyse the EKB and DKB for AIME. Based on the analysis we introduce a solution for each knowledge barrier and combine them into AIME-NoB, where NoB stands for **No B**arriers. The general framework of the solutions is shown in Figure 1 and the pseudocode of AIME-NoB is provided in Algorithm 1. Before diving into the analysis, we first formally define EKB and DKB with

$$\text{EKB} = \mathcal{R}(\pi_{\omega^*}) - \mathcal{R}(\pi_{\psi^*}), \tag{4}$$

$$\text{DKB} = \mathcal{R}(\pi_{\text{demo}}) - \mathcal{R}(\pi_{\omega^*}). \tag{5}$$

Let $\mathcal{D} := (D_{\text{demo}}, D_{\text{body}}, D_{\text{online}})$, then

- $\omega^* = \text{argmax}_\omega\, J_{\text{policy}}(\hat{p}_{\pi_{\text{demo}}}(a_t|o_{1:T}), \pi_\omega(a_t|o_{\leq t}), \mathcal{D})$ represents the optimal policy parameters for maximising $J_{\text{policy}}$ with the oracle.

- $\psi^* = \text{argmax}_\psi\, J_{\text{policy}}(q_{\phi^*}(a_t|o_{1:T}), \pi_\psi(a_t|o_{\leq t}), \mathcal{D})$ represents the optimal policy parameters for maximising $J_{\text{policy}}$ with the learned model.

- $J_{\text{policy}}(q(a_t|o_{1:T}), \pi(a_t|o_{\leq t}), \mathcal{D})$ is the learning objective for the policy depending on an action-inference model $q(a_t|o_{1:T})$. For behaviour cloning based methods like BCO and AIME, it is essentially equivalent to $-\sum_{o_{1:T}\in D_{\text{demo}}}\sum_t D_{\text{KL}}(q(a_t|o_{1:T})\,|\,\pi(a_t|o_{\leq t}))$.

- $\phi^* = \text{argmax}_\phi\, J_{\text{model}}(D_{\text{body}}, D_{\text{online}}, \phi)$ is the optimal parameter for maximising the model-learning objective $J_{\text{model}}$.

- $\hat{p}_{\pi_{\text{demo}}}(a_t|o_{1:T})$ is the ground truth of empirical distribution of the demonstration data that serves as an oracle.

Based on the definitions, we can clearly see that EKB is caused by the imperfect inference of the unknown actions of the observation-only demonstrations, which can be characterized by $D_{\text{KL}}(\hat{p}_{\pi_{\text{demo}}}(a_{1:T-1}|o_{1:T}) \,|\, q_{\phi^*}(a_{1:T-1}|o_{1:T}))$; while DKB is caused by the limited learning signal that $J_{\text{policy}}$ offers as it is only defined on the data points in the demonstration dataset. In Appendix J, we propose some upper bounds for both EKB and DKB based on these insights. In the rest of this section, we will focus on practical solutions for both of the barriers.

## 3.1 Overcoming the EKB

To reduce the EKB, we need to bring the learned inference model closer to the oracle, i.e. to minimise $\sum_{o_{1:T} \in D_{\text{demo}}} D_{\text{KL}}(\hat{p}_{\pi_{\text{demo}}}(a_{1:T-1}|o_{1:T}) \,|\, q_{\phi^*}(a_{1:T-1}|o_{1:T}))$. The most natural way to do so is to allow the agent to further interact with the environment, similar to how we humans practice for a novel skill. New experiences can minimise the error in the pretrained model near the policy $\pi_\psi$ and enhance task-specific embodiment knowledge. Torabi et al. (2018) proposed a modified version of BCO(0) called BCO($\alpha$) introducing such an interaction phase. However, empirical results show it did not overcome the EKB as a gap remains with the BC oracle when the environment is complex. In fact, as we will show in the following, the idea of adding online interactions is not straightforward to successfully implement in practice.

As shown in recent works in Offline RL, continuing training of an actor-critic from the offline phase in the online phase requires certain measures to combat the shift of objective (Lee et al., 2022; Ball et al., 2023; Nakamoto et al., 2023). A similar story also applies when extending AIME from purely offline to online. The most dominant problem we found is overfitting to the newly collected dataset.

As the training progresses alternating between data collection, model training and policy training, in the early phase of training there are only very few new trajectories available for training the model. Because the world model is highly expressive, it may overly favour similar trajectories, especially the action sequence, in the new data, leading to a high ELBO. Normally, this may not be a big problem since, eventually, more and more data will be collected to combat this overfitting. But since AIME also depends on the ELBO to train the policy, it quickly causes the policy training to diverge. That is to say, when the model is extensively trained on a small amount of data, it not only maximises the conditional likelihood $\log(o_{1:T}|a_{0:T-1})$ but also maximises the marginal likelihood $\log(\cdot|a_{0:T-1})$, which diverges the likelihood-based action inference process.

In order to address the overfitting issue, we need a regulariser for model learning. Instead of designing ad-hoc methods to regularise the model in the parameter space, we adopt a data-driven approach. From the model's perspective, the overfitting is caused by a sudden shift of the training data from a large and diverse pretraining dataset to a small and narrow replay buffer. To avoid this sudden shift, the most straightforward method is to just append new data to a replay buffer that is pre-filled with the pretraining dataset. However, this causes data efficiency problems since the newly collected data is relatively little compared to the big pretraining dataset. Uniformly sampling from the joint replay buffer hence overly limits usage of the new data. Instead, we suggest sampling separately from both datasets. We modify the model-learning objective $J_{\text{model}}$ in Equation (2) to

$$\phi^*, \theta^* = \underset{\phi, \theta}{\operatorname{argmax}} \; \alpha \mathbb{E}_{\{o,a\} \sim D_{\text{body}}, s \sim q_\phi}[\text{ELBO}] + (1 - \alpha)\mathbb{E}_{\{o,a\} \sim D_{\text{online}}, s \sim q_\phi}[\text{ELBO}]. \tag{6}$$

The amount of data we sample from the pretraining dataset is controlled by a hyper-parameter $\alpha$, which represents how much regularisation we put upon the model. Here we mainly consider setting $\alpha = 0.5$, so that we sample the data evenly from both datasets.

This finding contradicts Rajeswar et al. (2023) and Hansen et al. (2023a), where the pretrained world models do not need such a data-driven regulariser. We conjecture that unlike AIME, these approaches mainly use their world models purely as generative models to predict states and rewards given action sequences, which is only indirectly influenced by overfitting the ELBO.

---

**Algorithm 1** AIME-NoB

---

1: **Input:** Embodiment dataset $D_{\text{body}}$, Demonstration dataset $D_{\text{demo}}$, Pretrained world model parameters $\phi, \theta$, Surrogate reward model $R_\nu$, Regulariser ratio $\alpha$, Value gradient weight $\beta$, Batch size B
2: Initialise policy and critic parameters $\psi, \xi$ randomly.
3: **for** $i = 1$ **to** policy pretraining iterations **do**
4:     Draw a batch of demonstrations $o_{1:T} \sim D_{\text{demo}}$.
5:     Update policy parameters $\psi$ with Equation (3).
6: **end for**
7: Initialize $D_{\text{online}} \rightarrow \emptyset$.
8: **for** $i = 1$ **to** Environment Interaction budget **do**
9:     Collect a new episode $\{o_{1:T}, a_{1:T}\}$ with the current policy $\pi_\psi$
10:     Update the surrogate reward model $R_\nu$ if needed, e.g. for AIL.
11:     Estimate reward using surrogate reward model $r_{1:T}^{\text{sur}} = R_\nu(o_{1:T})$
12:     Append $\{o_{1:T}, a_{1:T}, r_{1:T}^{\text{sur}}\}$ to $D_{\text{online}}$
13:     *# Update world model*
14:     Draw $\alpha \cdot$ B samples $b_{\text{body}} \sim D_{\text{body}}$
15:     Draw $(1 - \alpha) \cdot$ B samples $b_{\text{online}} \sim D_{\text{online}}$
16:     Define combined batch $b = b_{\text{body}} \cup b_{\text{online}}$
17:     Finetune model with batch $b$ using Equation (6).
18:     *# Update policy*
19:     Sample a batch from $D_{\text{demo}}$
20:     Update policy parameters $\psi$ with Equation (9).
21:     Update value function parameters $\xi$ with Equation (8).
22: **end for**

---

## 3.2 Overcoming the DKB

Based on the discussion from the previous sections, the straightforward way of overcoming the DKB is also to increase the number of demonstrations available to the agent. However, expert demonstrations are difficult and expensive to collect. Increasing the size of the demonstration dataset is not always feasible in real-world applications. Alternatively, we can modify the policy learning objective $J_{\text{policy}}$ in Equation (3) to reduce the generalisation gap. In order to propose a practical solution, we first need to look deeper into what is the real cause of the DKB.

The policy-learning part of the AIME algorithm is essentially behaviour cloning, and it is only conducted on the demonstration dataset. So for the states covered in the demonstration dataset, the policy is given clear guidance about what to do, while for other states, the behaviour is undefined. AIME solely relies on the generalisation abilities of the learned latent state and the trained policy network to extrapolate the correct behaviour. In particular for small demonstration datasets, this can be unreliable or even impossible. Therefore, if we were able to enlarge the space of the covered states, we should reduce the DKB (Ross et al., 2011).

Based on these insights, we propose to introduce a surrogate reward providing a guiding signal for the agent on the replay buffer dataset, i.e. $r_{0:T}^{\text{sur}} = R_\nu(o_{0:T})$. Using this reward, we train the policy with a dreamer-style actor-critic algorithm based on imagination in the latent space of the world model (Hafner et al., 2019a). In order to do this, we first need to modify the reconstruction term in Equation (1) by adding an extra term for decoding the surrogate reward, i.e. $\log p_\theta(r_t^{\text{sur}}|s_t)$. Then, we further train a value estimator $V_\xi(s_t)$ using TD($\lambda$)-return estimates, i.e.

$$V_\xi^\lambda(s_t) = (1 - \lambda) \sum_{n=1}^\infty \lambda^{n-1} V_\xi^{(n)}(s_t) \tag{7}$$

$$\text{with } V_\xi^{(n)}(s_t) = \sum_{t'=t+1}^{t+n} \gamma^{t'-t-1} r_{t'}^{\text{sur}} + \gamma^n V_\xi(s_{t+n}).$$

Using this estimate, we optimise our value function by minimising the MSE, i.e.

$$\xi^* = \operatorname*{argmin}_{\xi}(V_\xi(s_t) - V_{\xi'}^\lambda(s_t))^2. \tag{8}$$

As is common practice, we use a target value network with parameters $\xi'$ to stabilise training, whose parameters are updated using Polyak averaging with a learning rate $\tau$ in every iteration.

Using this value estimate, we extend the policy-learning objective $J_{\text{policy}}$ of Equation (3) to

$$\psi^* = \operatorname*{argmax}_{\psi} \mathbb{E}_{o\sim D_{\text{demo}},s\sim q_{\phi,\theta},a\sim\pi_\psi}[\text{ELBO}] + \beta\mathbb{E}_{\{o,a\}\sim D_{\text{online}},s\sim q_\phi,a'\sim\pi_\psi,s'\sim p_\theta}[V_{\xi'}^\lambda(s')], \tag{9}$$

where $\beta$ is a hyper-parameter for balancing the two terms. We set $\beta = 1.0$ by default in this paper.

There could be many choices to derive this surrogate reward model. In this paper, we consider three different types of surrogate reward, namely AIL, OT and VIPER. AIL (Ho & Ermon, 2016; Torabi et al., 2019) uses adversarial training to learn a discriminator to tell whether an observations is generated by the expert, and uses the score from the discriminator as the reward. OT (Papagiannis & Li, 2023; Haldar et al., 2022) uses optimal transport theory to measure the distance between a given trajectory and a group of expert trajectories, and uses the negative distance as the reward. VIPER (Escontrela et al., 2023) learns a video prediction model from the demonstration datasets and uses the likelihood from the trained model as the reward. The detailed explanation of the three variants are in Appendix A. We use AIL as the default variant for AIME-NoB.

## 4 Experiments

In the experiments, we aim to answer the following questions: **Q1:** How does the proposed AIME-NoB compare with state-of-the-art methods on common benchmarks? **Q2:** How well do the proposed modifications overcome the EKB and the DKB? **Q3:** How do different design choices, components and hyper-parameters influence the results? In order to answer these questions, we design our experiments on DMC and MetaWorld benchmarks. The main results and the setup are described in the following sections, while some additional interesting results are presents in Appendix I.

### 4.1 Datasets and Tasks

For the DMC benchmark, we choose nine tasks across six embodiments following Liu et al. (2022a) and use the same published dataset (Haldar et al., 2022) as the demonstration datasets. Each dataset contains only 10 trajectories to reflect the scarcity of expert demonstrations. For the embodiment dataset, in order not to leak the task information from the pretraining phase, we follow Rajeswar et al. (2023) and run a Plan2Explore (Sekar et al., 2020) agent for each embodiment with 2M environments steps and use its replay buffer as the embodiment dataset. Different to them taking the model directly from the Plan2Explore agent as the pretrained model, we follow Zhang et al. (2023) to retrain the model for 200k gradient steps to get a better model. When evaluating the performance of the learned policy on each task, we rollout the policy 10 times with the environment, and report the mean return.

For vision-based MetaWorld benchmark, we use the data from Hansen et al. (2023a). The embodiment dataset was created from the replay buffer datasets. The open-sourced replay buffer datasets contain 40k trajectories for each of the 50 tasks with only state information. In order to fit to our image observation setup, we render the images by resetting the environment to the initial state of each trajectory and then executing the action sequence. The details can be found in Appendix E.

With respect to the embodiment dataset, following the idea of not leaking too much about the task information, inspired by the common practice in offline RL benchmarks (Fu et al., 2021), we use the first 200 trajectories from each replay buffer and form a dataset with 10k trajectories in total. We call this dataset MW-mt50 and we use it for the benchmark on MetaWorld to compare AIME-NoB with other algorithms. To further study the out-of-distribution transfer ability of the pretrained model, we follow the difficulty classification of the tasks from (Seo et al., 2022a) and only use the 39 easy and medium difficulty tasks to generate the datasets

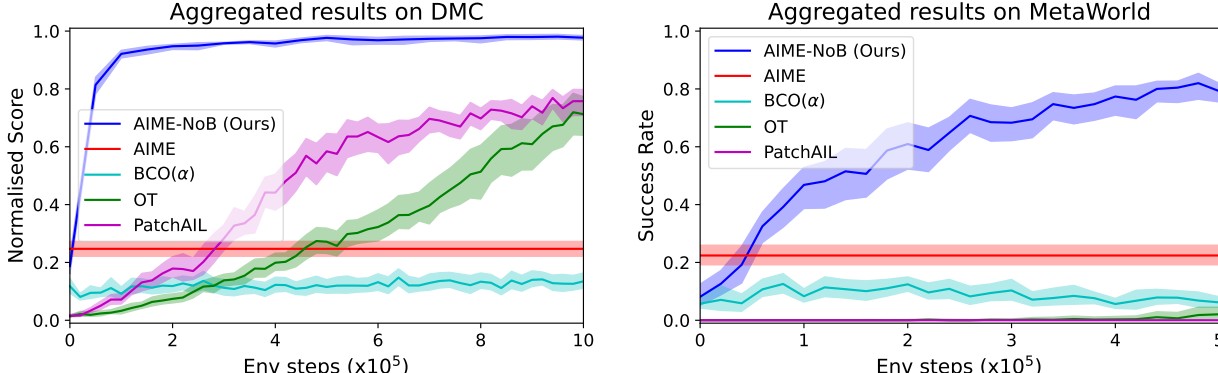

Figure 2: Comparing AIME-NoB with other algorithms. The figures show aggregated IQM scores on 9 DMC tasks and 6 MetaWorld tasks. All the algorithms are evaluated with 5 seeds on each task and the shaded region representing 95% CI.

and use the 11 hard and very hard tasks as hold-out tasks. We uniformly sample 250 trajectories from the first 10k trajectories from each of the 39 tasks and form a dataset with 9750 trajectories in total. We refer to this dataset as MW-mt39.

For evaluating the algorithms, we choose four hard or very hard tasks, namely disassemble, assembly, hand-insert and push; and two medium difficult tasks, namely sweep and hammer. As for the demonstration datasets, we use the single-task policies open-sourced by TD-MPC2 and collect 50 trajectories for each task. We ensure that every trajectory in the demonstration dataset is successful. For evaluation on each MetaWorld task, due to the noisy nature of the task, we rollout the policy 100 times with the environments, and report success only if the very last time step of an episode is marked as successful by the environment (following Hansen et al. (2023a)).

### 4.2 Implementation

For the world model, we use the RSSM architecture (Hafner et al., 2019b) with the hyper-parameters in Hafner et al. (2019a) for DMC tasks. In addition, we use the KL Balancing trick from Hafner et al. (2020) to make the training more stable. For MetaWorld, since the visual scene is more complex, we use the M size model from Hafner et al. (2023), but still with the continuous latent variable to be aligned with other models used in this paper. The policy network is implemented with a two-layer MLP, with 128 neurons for each hidden layer. All the models are trained with Adam optimiser (Kingma & Ba, 2017). More details on hyper-parameters are in Appendix C.

### 4.3 Results

**Q1: How does AIME-NoB compare with state-of-the-art methods on common benchmarks?**
We compare the AIL variant of AIME-NoB with the representative state-of-the-art algorithms:

- AIME (Zhang et al., 2023) represents the base algorithm that we improve upon. The algorithm uses pretrained world model offline, and suffers from both EKB and DKB.

- BCO($\alpha$) (Torabi et al., 2018) is an online extension of the popular BCO(0) algorithm, which represents another line of methods that can use pretrained models. The online interactions in BCO($\alpha$) can potentially overcome EKB from BCO(0).

- PatchAIL (Liu et al., 2022a) represents the Generative Adversarial Imitation Learning (GAIL) styles of algorithms (Ho & Ermon, 2016; Torabi et al., 2019).

- OT (Haldar et al., 2022) represents the trajectory matching based algorithms.

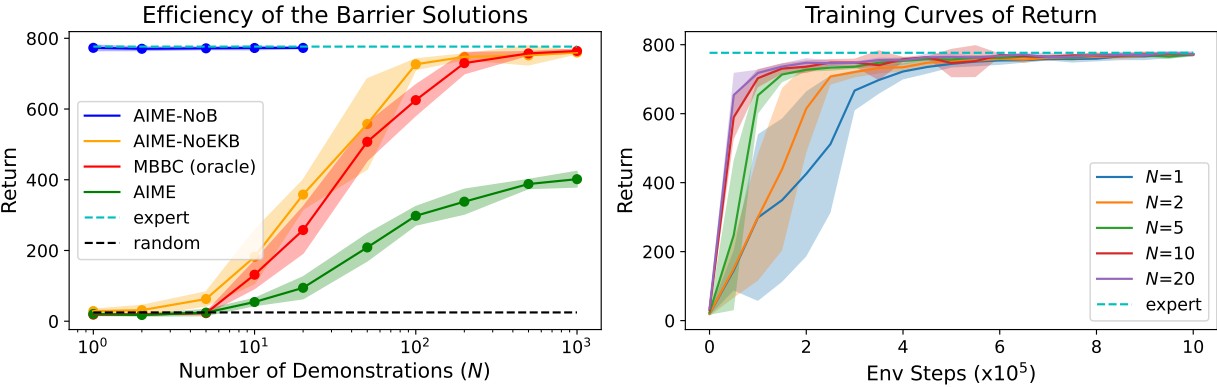

Figure 3: Performance of AIME-NoB, AIME-NoEKB, MBBC, AIME w.r.t. different number of demonstrations on the walker-run task. For AIME-NoB, we do not show the result for more than 20 demonstrations since it is already saturated to the expert. All results are averaged across 5 seeds with the shaded region representing a 95% CI.

The results of the benchmark are shown in Figure 2. We follow Agarwal et al. (2022) to report the IQM score and 95% CI when aggregating over all the tasks in the same suite. For each DMC task, the score is normalised using the average return of the expert, while for each MetaWorld task we directly report the success rate. From the results, we can clearly see AIME-NoB achieves better sample-efficiency and final performance on both of the benchmark suites comparing with all other algorithms. Benefiting from the pretrained world model, AIME-NoB typically can reach near expert performance within 100k environment steps on DMC tasks. This even matches the performance in Rajeswar et al. (2023) where the true rewards are available. For the complete results of each individual task please refer to Appendix H.

**Q2: How well do the proposed methods overcome knowledge barriers?** In order to show how well AIME-NoB overcomes the two knowledge barriers, we conduct the same experiment as in Figure 1 by providing the agent with different numbers of demonstrations on walker-run. We also run an additional variant coined AIME-NoEKB where we only apply the solution for the EKB. The result is shown in Figure 3. As we discussed before, MBBC as an oracle method that circumvents the EKB is an upper bound for AIME. AIME-NoEKB matches and even slightly outperforms MBBC, which implies the proposed solution completely overcomes the EKB. The fact that it slightly outperforms MBBC is a bonus of the model choice – fine-tuning the latent variable world model improves the generalization of the latent space which mitigates the DKB. AIME-NoB, which further addresses the DKB, matches the expert performance even when given only 1 demonstration. This showcases that the DKB has also been completely overcome using the surrogate reward. The difference between different number of demonstrations is mainly on the sample-efficiency side, where the more demonstrations we have the less online interactions we need to attain the expert performance.

**Q3.1: Which variant of AIME-NoB performs the best?** We run all the three AIME-NoB variants with different surrogate rewards, i.e. AIL, OT and VIPER, on the two benchmark suites and aggregate the results. From Figure 4a and 4b, we can see the AIL variant of AIME-NoB generally performs the best on both suites. On the DMC suite, all the three variants managed to converge to the same final performance, but the AIL variant is slightly more sample efficient. On the MetaWorld suite, the distinction is larger between different variants, which highlights the supremacy of the AIL variant. We hypothesise that AIL is the only variant that directly adapts the surrogate rewards online by training the discriminator. For the other two variants: VIPER has zero adaptivity since the VIPER model is fixed during the online training; OT has an indirect adaptivity since it depends on the image encoder from the world model which get finetuned during online learning. This adaptivity enables AIL to capture more pertinent signals during the imitation process. Based on these results, we choose the AIL variant as the default implementation for AIME-NoB.

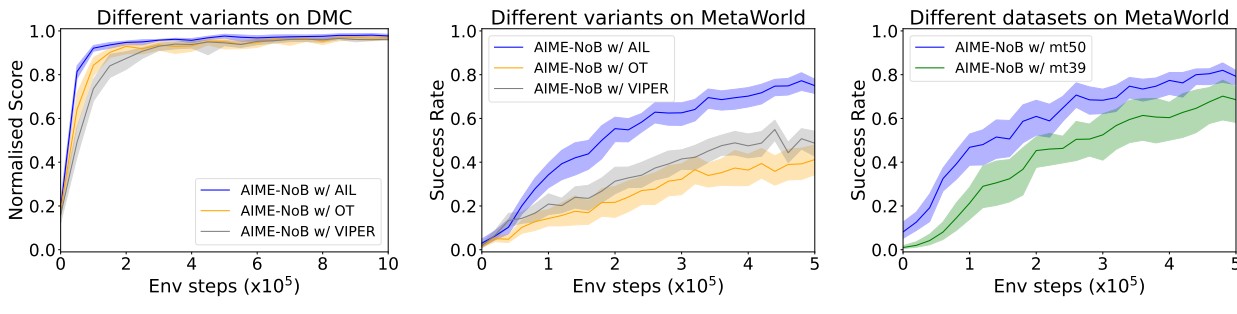

(a) Different variants on DMC.        (b) Different variants on MetaWorld.        (c) Different datasets on MetaWorld.

Figure 4: Ablations of different variants of AIME-NoB and choices of the embodiment datasets. All the algorithms are evaluated with 5 seeds with the shaded region representing 95% CI.

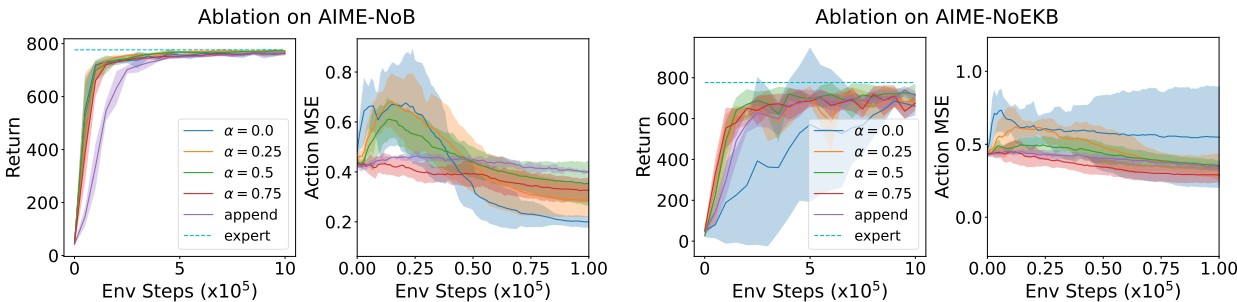

Figure 5: Ablations of replay ratio $\alpha$ on walker-run task. AIME-NoB is running with 10 demonstrations, while AIME-NoEKB is with 100 demonstrations. Action MSE is only shown for the first $10^5$ env steps. All results are averaged across 5 seeds with the shaded region representing a 95% CI.

**Q3.2: Which dataset can pretrain the better world model for AIME-NoB?** The quality of the pretrained models naturally depends on the quality of the pretraining datasets. In order to understand what characteristics of the datasets influence the performance, we train all the AIME-NoB variants with two different models pretrained separately on the MW-mt39 datasets and MW-mt50 datasets. The aggregated result with each dataset is shown in Figure 4c. The result demonstrates that although the size of the datasets is roughly the same, the model pretrained on MW-mt50 offers better results. This may imply covering diverse behaviours and objects is more valuable than knowing the expert, for example the mt50 dataset contains objects to be assembled while the mt39 does not.

**Q3.3: How does different data regulariser ratio $\alpha$ influence the performance?** We ablate the regulariser ratio $\alpha$ from [0.0, 0.25, 0.5, 0.75]. Further, we compare to a simple *append* version where the online dataset is appended to the embodiment dataset and treated as a singular dataset for sampling. The append version can be also understood in this experiment as having an inverse proportional schedule of $\alpha$ from 1.0 to 0.66 during the course of training. To isolate the effect on the EKB, we train both AIME-NoB and AIME-NoEKB. We show both the training curves of the return and action MSE, which is the MSE between the inferred actions and the true actions, in Figure 5. For AIME-NoEKB, as long as we enable the regulariser, i.e. set $\alpha > 0$, we get reliable improvements of returns over the course of training. But if we disable the regulariser by setting $\alpha = 0$, the return exhibits high variance. The cause is clear by looking at the action MSE. For $\alpha = 0$ the action MSE diverges in the beginning and cannot recover. For AIME-NoB, the story is more complicated. While the action MSE still diverges in the beginning when $\alpha \leq 0.5$, the surrogate reward can guide the policy back on track and even achieves lower action MSE after recovery. In this case, a small $\alpha$ helps the algorithm make more use of the online dataset, resulting in higher sample efficiency. Although we keep the choice of $\alpha$ with a fixed value for AIME-NoB in the light of simplicity, there should be

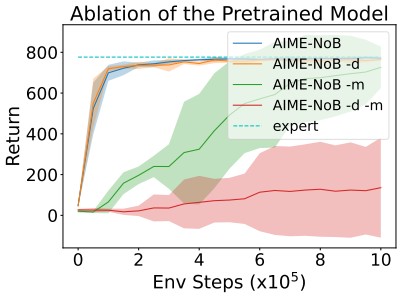 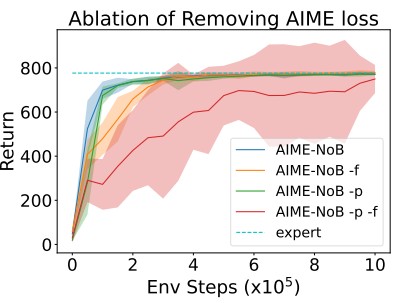 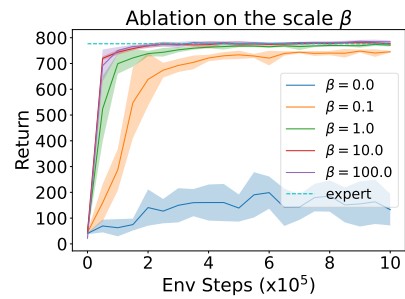

(a) Effect of the pretrained world model. -m means removing the pretrained world model while -d means removing the pretrained dataset.

(b) Ablation of removing AIME loss from AIME-NoB. -p means removing from pretraining while -f means removing from finetuning.

(c) Ablation for the choice of the weight of the value gradient loss $\beta$.

Figure 6: Ablations of pretrained models, AIME loss and the weight of the value gradient loss $\beta$. All the algorithms are evaluated with 5 seeds with the shaded region representing 95% CI.

a smart scheduling for each task to strike the balance between stability and sample-efficiency. However, we think finding this optimal strategy is beyond the scope of this paper.

**Q3.4: How much benefit do we get from the pretrained world model and the dataset for pretraining?** One of the advantages of AIME-NoB over popular ILfO algorithms is that it can make use of pretrained models and pre-collected datasets. Thus, we want to investigate how much AIME-NoB benefits from having a pretrained model and a pre-collected dataset. We rerun AIME-NoB on walker-run without the embodiment dataset and without the pretrained world model. As we can see the result from Figure 6a, without the pretrained the world model, the sample-efficiency is largely affected. Between the two components, the model is more important than the dataset.

**Q3.5: How do different values of the gradient loss weight $\beta$ influence the performance?** We set the weight $\beta$ from [0.0, 0.1, 1.0, 10.0, 100.0] and plot the results in Figure 6c. As the result shows, without the surrogate reward, i.e. $\beta = 0$, the agent cannot reach expert performance due to the DKB. Having a small $\beta$ slows learning progress toward convergence. On the other hand, setting $\beta$ to a much larger value will improve the sample-efficiency without causing instability. For the sample efficiency, since we only have 10 demonstrations, the DKB dominates over the EKB as shown in Figure 3. Thus, having a larger $\beta$ will speed up learning. In terms of stability, as we discussed in 3.2, AIME loss and the value gradient loss operate on different regions of the environment states. This could make their influence on the policy independent of each other.

**Q3.6: Is surrogate reward all you need?** Given the results above that AIME-NoB can work well even when we lower the effect of the AIME loss by making either worse action inference, i.e. set $\alpha$ to low value, or strengthen the value gradient loss, i.e. set $\beta$ to high value, a natural question to ask is whether the AIME loss is still needed in AIME-NoB or whether surrogate reward only is enough to solve the imitation learning task. The AIME loss is used in two places in the AIME-NoB algorithm – both for pretraining the policy offline and finetuning the policy online. We compare AIME-NoB with variants that removes AIME from these parts. Form the results shown in Figure 6b, removing AIME loss from either pretraining or finetuning will lower the sample efficiency. Removing from both phases causes instability and convergence issues during training. Thus, AIME loss is still crucial and cannot simply be replaced by surrogate rewards.

# 5 Related Work

**Imitation Learning from Observation.** ILfO (Torabi et al., 2018; 2019; DeMoss et al., 2023; Li et al., 2023; Baker et al., 2022; Zhang et al., 2023; Liu et al., 2022a) becomes more popular in recent years due to their potential to utilise internet-scale videos for behaviour learning. Most of the previous works (Torabi et al., 2018; 2019; Li et al., 2023; Kidambi et al., 2021) study the problem only with the true state as observation.

Recent works (DeMoss et al., 2023; Baker et al., 2022; Zhang et al., 2023; Liu et al., 2022a) have started to shift toward image observations as a more general setting. Only few works (Zhang et al., 2023; Torabi et al., 2018) can leverage pretrained models. Our work is a continuation of this journey and further emphasise the performance benefit from pretrained models.

**Pretrained Models for Decision-Making.** Inspired by the tremendous progress made in recent years in CV and NLP fields with the power of pretrained models, the decision-making community is also trying to follow the trend. Most recent works focus on the use of Large Language Model (LLM) for decision-making. A prompted model is used for producing trajectories and plans (Chen et al., 2024; Huang et al., 2022; Ahn et al., 2022; Di Palo et al., 2023), code (Vemprala et al., 2023; Liang et al., 2023; Singh et al., 2022; Chen et al., 2023; Huang et al., 2023) or for modifying the reward (Ma et al., 2023; Mahmoudieh et al., 2022). There are also other people studying the benefit of pretrained visual models for visuomotor tasks (Shah & Kumar, 2021; Majumdar et al., 2023; Hansen et al., 2023b; Parisi et al., 2022) while others try to train large policy networks directly with transformers (Vaswani et al., 2017) and huge datasets (Brohan et al., 2023; Brohan et al.; Reed et al., 2022). However, there is only little attention being put on pretrained world models (Zhang et al., 2023; Rajeswar et al., 2023; Sekar et al., 2020), which are natively developed by the model-based decision-making community and perfectly fit into the pretraining and finetuning paradigm. Our work explores this overlooked domain and showcases its potential.

## 6 Discussion

In this paper, we identify two knowledge barriers, namely the EKB and the DKB, which limit the performance of state-of-the-art ILfO methods using pretrained models. We thoroughly analyse the underlying cause of each barrier and propose practical solutions. Specifically, we propose to use online interaction with a data-driven regulariser to overcome the EKB and surrogate reward labelling to reduce the DKB. Combining these solutions, we propose AIME-NoB and showcase its supreme efficiency compared to SOTA ILfO methods. Our ablation studies show how each knowledge barrier is addressed by the proposed solution and how different design choices influence the performance.

However, there are still limitations for the scope of this study. First, this work is mainly an empirical study. Some theoretical results could enhance the understanding of the knowledge barriers. Second, as suggested by the ablation study, a careful design for a schedule of $\alpha$ could further improve the sample efficiency. Third, due to the high demand of computing resources, we only study the pretrained world model on a rather small scale, i.e. the biggest model has only 20M parameters. It will be interesting to study how well the algorithm scales to larger models. Lastly, although we have shown a single pretrained world model can be used for multiple tasks, the power of pretrained world models are not fully realized. It would be interesting to see how world models can be used to train a multi-tasks policy. These limitations provide directions for future works.

We hope our work can shed some light on the future development of ILfO method and bring more attention to the great potential of pretrained world models.

### Acknowledgments

We would like to thank Botond Cseke for his help in reviewing the mathematical aspects of this paper.

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

## A Implementation details of AIME-NoB variants

**AIME-NoB with AIL rewards** The idea of AIL (Ho & Ermon, 2016; Torabi et al., 2019) is to train a discriminator $D_\nu$ to classify whether the observation is from the expert demonstration or the agent's rollout. As the observation is an image in our case, we follow previous works (Haldar et al., 2022; Liu et al., 2022a) to apply the discriminator upon the feature of an image encoder, for which we naturally reuse the image encoder $f_\phi$ from the world model. In this way, the discriminator is trained by

$$\nu^* = \underset{\nu}{\arg\min} \ \mathbb{E}_{o^- \sim D_{\text{body}} \cup D_{\text{online}}, o^+ \sim D_{\text{demo}}, \alpha \sim U(0,1)}$$

$$[D_\nu(f_\phi(o^-)) - D_\nu(f_\phi(o^+)) + \lambda ||\nabla_\nu D_\nu(\alpha f_\phi(o^-) + (1-\alpha)f_\phi(o^+)||_2^2]. \tag{10}$$

The first two terms in Equation (10) train the discriminator to put higher score on the observations from the demonstrations while lower score on the observations generated by the agent. The last term is a regulariser to smooth out the landscape of discriminator. The regulariser weight $\lambda$ is set to 10. Following previvous works (Haldar et al., 2022; Liu et al., 2022a), the discriminator is implemented as a 3-layers MLP with 1024 hidden units for each layer and trained with Adam optimiser with learning rate of 1e-4. With a trained discriminator, the AIL reward can be computed as

$$r_T^{\text{AIL}} = \log D_\nu(f_\phi(o_T)) - \log(1 - D_\nu(f_\phi(o_T))). \tag{11}$$

Since the value of AIL rewards also depends on the image encoder $f_\phi$, which is changing during the course of training, previous works (Liu et al., 2022a; Haldar et al., 2022) maintain a slow-update version of the image encoder to mitigate the non-stationary of the reward. But since our image encoder is pretrained as part of the world model and finetuned with the data-driven regulariser, we find its weight is stable during the course of training. Thus, we don't apply this additional slow encoder. In the experiments, we find the adversarial training is stable with the help of the world model, as also shown in Rafailov et al. (2022).

**AIME-NoB with OT rewards** OT (Papagiannis & Li, 2023; Haldar et al., 2022) was introduced to imitation learning to alleviate the non-stationary reward and sensitive to hyper-parameters problems in AIL. The idea is to use the optimal transport to measure the minimal effort of moving any trajectory $\mathcal{T} = \{o_{1:T}\}$ to a demonstration trajectory $\mathcal{T}^d = \{o^d_{1:T^d}\}$, where the length of the trajectory $T$ and $T_d$ can be different. The effort of moving the trajectory is measure by a Wasserstein distance, i.e.

$$g(\mu, f_\phi(\mathcal{T}), f_\phi(\mathcal{T}^d), c) = \sum_{t=1}^{T} \sum_{t'=1}^{T^d} \mu_{t,t'} c(f_\phi(o_t), f_\phi(o^d_{t'})), \tag{12}$$

where $\mu \in \mathbb{R}^{T \times T^d}$ is the transportation matrix that satisfy $\mu\mathbf{1} = \frac{1}{T}\mathbf{1}$ and $\mu^T\mathbf{1} = \frac{1}{T^d}\mathbf{1}$, $f_\phi$ is the process function that map the raw observation to a metric space for which we reuse the image encoder as in AIL, $c$ is the cost function that measure the distance between two vectors in the metric space for which we use the cosine distance. With the cost measure $g$, the optimal transport solve for the optimal transportation matrix with

$$\mu^* = \underset{\psi}{\arg\min}\, g(\mu, f_\phi(\mathcal{T}), f_\phi(\mathcal{T}^d), c). \tag{13}$$

With the optimal transportation matrix, the OT reward can be defined as

$$r^{\text{OT}}_T = -\lambda \sum_{t=1}^{T^d} \mu^*_{T,t} c(f_\phi(o_T), f_\phi(o^d_t)). \tag{14}$$

The $\lambda$ is hyper-parameter to scale the OT reward to be easier for the agent to learn with. We follow Haldar et al. (2022) to apply an adapted normalisation scheme where the scale factor is based on the cost measure of the first trajectory we evaluate, i.e.

$$\lambda = \frac{4}{g(\mu^*, f_\phi(\mathcal{T}^{(1)}), f_\phi(\mathcal{T}^d), c)}. \tag{15}$$

When multiple demonstrations are available, we take the OT reward from the trajectory with the lowest total transportation cost.

It may worth note that OT is the only variant that doesn't requires to train a separate model for the reward labelling. However, it is still changing during the course of training since the image encoder $f_\phi$ gets finetuned.

**AIME-NoB with VIPER rewards** VIPER (Escontrela et al., 2023) trains a video prediction model on the demonstration datasets and treats the likelihood of the video prediction model as the reward for policy learning, i.e.

$$r^{\text{VIPER}}_T = \log p_\nu(o_T | o_{t<T}). \tag{16}$$

In the original paper, the authors first pretrain a VQ-GAN (Esser et al., 2021) from a multi-tasks expert dataset, and then train a GPT-style auto-regressive model in the quantised space for prediction. For a fair comparison with other variant, we consider to only train the VIPER model on the single demonstration dataset for the task. For simplicity of the implementation, in this paper, we consider training an unconditioned latent world model as in Seo et al. (2022b) to model the VIPER reward. We use the same RSSM architecture of the model learning for DMC, only removing the condition of the actions, and we train the VIPER model for each task separately. Especially during training, we find training such a powerful model from scratch

on a small dataset can easily result in over-fitting. Thus, we empirically choose to train the model only for 500 gradient steps for DMC models and 1000 gradient steps for MetaWorld models. We show evidence of overfitting in Appendix G.1. Due to the large scale of the ELBO, we also apply symlog (Hafner et al., 2023) when computing the VIPER reward. Another difference with the original VIPER paper is that we do not use intrinsic motivation as the exploration bonus as the authors suggested, since the AIME loss for policy learning already provides task-related guidance for exploration. We only apply an entropy regulariser to the policy as is common practice. We further show the synergy between AIME and VIPER in Appendix I.

## B  Compute Resources

All the experiments are run on a local cluster with a few A100 and RTX8000 instances. All the experiments are tuned to use less than 10GB of GPU memory so that they can run in A100 MIG. World models pretraining requires about 24 GPU hours, while VIPER models require negligible time for training. Each DMC experiment requires about 40 GPU hours while each MetaWorld experiment requires about 20 GPU hours. And for reference, the baseline algorithm AIME requires about 7 GPU hours for each task.

## C  Hyper-parameters

Here, we document the detailed hyper-parameters for all the trained models in Table 1.

## D  Source of Datasets

We use the expert trajectories from Haldar et al. (2022) at `https://osf.io/4w69f/?view_only=e29b9dc9ea474d038d533c2245754f0c`. The authors didn't provide a License for their dataset. Further, we use the replay buffer dataset from Hansen et al. (2023a) at `https://huggingface.co/datasets/nicklashansen/tdmpc2/tree/main/mt80`. The authors provide the dataset under the MIT License. Moreover, we use the replay buffer dataset from Zhang et al. (2023) at `https://github.com/argmax-ai/aime/tree/main/datasets`. The authors provide the dataset under the CC BY 4.0 License.

## E  Details for Resetting MetaWorld Tasks

To generate the image observation datasets from the TD-MPC2 replay buffer (Hansen et al., 2023a), we modify the MetaWorld codebase to reset the environment to the initial state of the trajectory from the first observation. Luckily, the starting position of the robot arm is always the same for each task, so that we do not need to apply inverse kinematics to solve for the initial pose of the robot arm. For the object and the target position, for most of the tasks, the internal reset position can be computed by making a constant shift on the object position and the target position in the observations. There are, however, also a few edge cases which we handle differently.

In button-press-topdown and button-press-topdown-wall, the object's true position only appears in the observation upon the second time step, presumably due to some simulator delay in the resetting process. So for these two tasks, the initial state is reset by the second observation.

For basketball and box-close, it seems like there is some internal collision detection that will alter the object and robot position after the task is reset, so computing the exact reset value from the observation is not possible. For these two tasks, we instead resort to a search-based method. To be specific, we use a gradient-free optimiser from (Liu et al., 2022b) to search over the resetting space of the object and find the reset position that minimises the L2 distance with the true observation.

More details of the implementation can be found in the code.

Table 1: AIME-NoB hyper-parameters use for each benchmark.

| | DMC | METAWORLD |
|---|---|---|
| **World Model** | | |
| CNN STRUCTURE | HA & SCHMIDHUBER (2018) | HAFNER ET AL. (2023) |
| CNN WIDTH | 32 | 48 |
| MLP HIDDEN SIZE | 512 | 640 |
| MLP HIDDEN LAYER | 2 | 3 |
| MLP ACTIVATIONS | LAYERNORM + SWISH | |
| DETERMINISTIC LATENT SIZE | 512 | 1024 |
| STOCHASTIC LATENT SIZE | 30 | |
| FREE NATS | 1.0 | |
| KL BALANCING | 0.8 | |
| MODEL SIZE | 8M | 20M |
| **Policy** | | |
| HIDDEN SIZE | 128 | |
| HIDDEN LAYER | 2 | |
| ACTIVATION | ELU | |
| DISTRIBUTION | TANH-GAUSSIAN | |
| **Value network** | | |
| HIDDEN SIZE | 128 | |
| HIDDEN LAYER | 2 | |
| ACTIVATION | ELU | |
| TARGET EMA DECAY | 0.95 | |
| **Training** | | |
| BATCH SIZE | 50 | 16 |
| HORIZON | 50 | 64 |
| TOTAL ENV STEPS | 1M | 500k |
| UPDATE RATIO | 0.1 | |
| GRADIENT CLIP | 100 | |
| POLICY ENTROPY REGULARISER WEIGHT | 1e-4 | |
| MODEL LEARNING RATE | 3e-4 | |
| POLICY LEARNING RATE | 3e-4 | |
| VALUE NETWORK LEARNING RATE | 8e-5 | |
| DISCOUNT FACTOR $\gamma$ | 0.99 | |
| TD-LAMBDA PARAMETER $\lambda$ | 0.95 | |
| IMAGINE HORIZON | 15 | |
| **AIME-NoB specific** | | |
| POLICY PRETRAINING ITERATIONS | 2000 | |
| DATA-DRIVEN REGULARISER RATIO $\alpha$ | 0.5 | |
| VALUE GRADIENT LOSS WEIGHT $\beta$ | 1.0 | |

## F Difference between compounding error and DKB

For DKB, most literature (Ho & Ermon, 2016; Peng et al., 2021; Torabi et al., 2019; Liu et al., 2022a) understand it as compounding error, in which the error is understood as when placing the policy with the **same** initial states as the demonstration, the learned policy will gradually diverge from the demonstrations due to error accumulation. Our DKB is a broader concept than compounding error. DKB attributes the gap of perform to the lack of learning signal on unsupported space of the demonstrations. In this way, EKB can explain more empirical successes in the literature than compounding error can. For example, a recent work MAHALO (Li et al., 2023) shows evidence of the importance of the size of the covered space. The

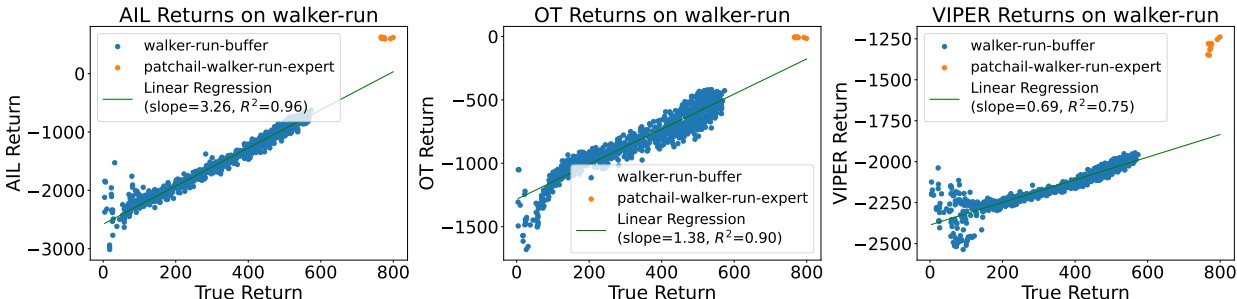

Figure 7: Correlation of surrogate rewards and true rewards on the walker-run task.

authors studied a similar ILfO setup with embodiment and demonstration datasets. They compared two variants: for one they train an Inverse Dynamics Model (IDM) from the embodiment dataset and use it to label the demonstration dataset, while for another they train a reward model from the demonstration dataset by labelling all time steps with a reward of 1, and then use it to label the embodiment dataset. Finally, they run the same offline RL algorithm on both labelled datasets. The results show the second variant attains a much better performance even though the labelling from the reward model is not as meaningful as the actions from the IDM.

# G  Landscape of the surrogate rewards

To better understand the different types of surrogate reward and why one works better than the others, we investigate the relevance between the surrogate reward and the true reward. We take the one seed of the final model from each of the variant on walker-run task and evaluate the surrogate rewards on both the expert dataset from PatchAIL, where the surrogate reward model is based on, and the replay buffer dataset from Zhang et al. (2023).

As we can see from the results in Figure 7, all the surrogate rewards manage to put the expert demonstrations with a higher reward than the trajectories in the replay buffer. Among them, AIL, which performs the best, has a more linear correlation with the true reward and a higher slow for the linear regression.

## G.1  Overfitting of the VIPER model

To better illustrate the overfitting problem for VIPER models and justify our choice of training fewer iterations, we train the VIPER models for a varying number of gradient steps and evaluate the correlations between the VIPER reward and the true reward. Specifically, we train the same VIPER model with {100, 500, 1000, 2000, 5000, 100000} gradient steps and plot the result in Figure 8. As we can clearly see, when training with less than or equal to 1000 gradient steps, VIPER reward has a very nice correlation with the true reward, with the middle-range performance even like a linear correlation. The best model could be selected from 500 and 1000 gradient steps. However, as we train the model for longer, the VIPER reward for the expert trajectories is boosted even higher, and as a side effect, it also relatively boosts up the VIPER reward for low-performance trajectories. This is because, when overfitting the expert trajectories, the model increases the marginal likelihood of all the observations in the expert trajectories to a higher value, which also includes a few frames of the robot lying on the ground at the very beginning of each trajectory after reset. For these low-performance trajectories, the robot remains mainly stuck around the initial position and struggles on the ground. This artifact of the overfitted VIPER reward creates a sharp local maximum in the low-performance region that the agent can hardly get away from.

# H  Full results of the benchmark

In this section, we show the full results of all the variants on the 15 tasks on the two benchmark suites. The aggregated results of all the variants are shown in Figure 9 and results on each individual tasks are shown in

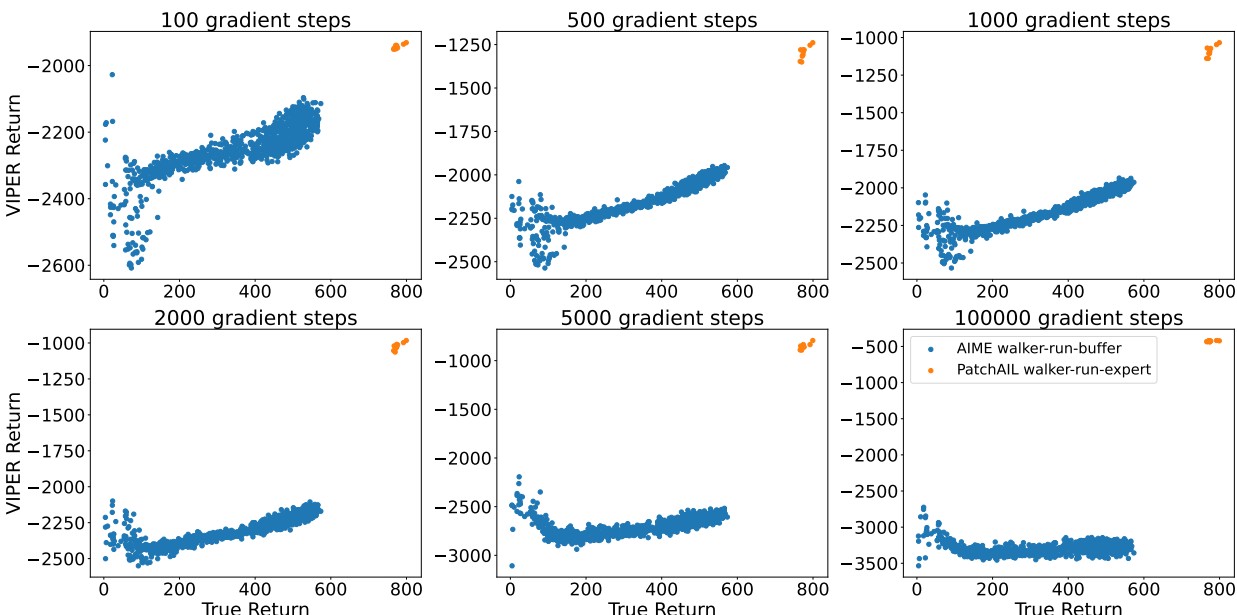

Figure 8: Correlation of the VIPER reward and the real reward with models trained with different numbers of gradient steps. Each point represents one trajectory. We can clearly see the model gradually overfitting and losing the correlation with the real reward when training for more than 1000 gradient steps.

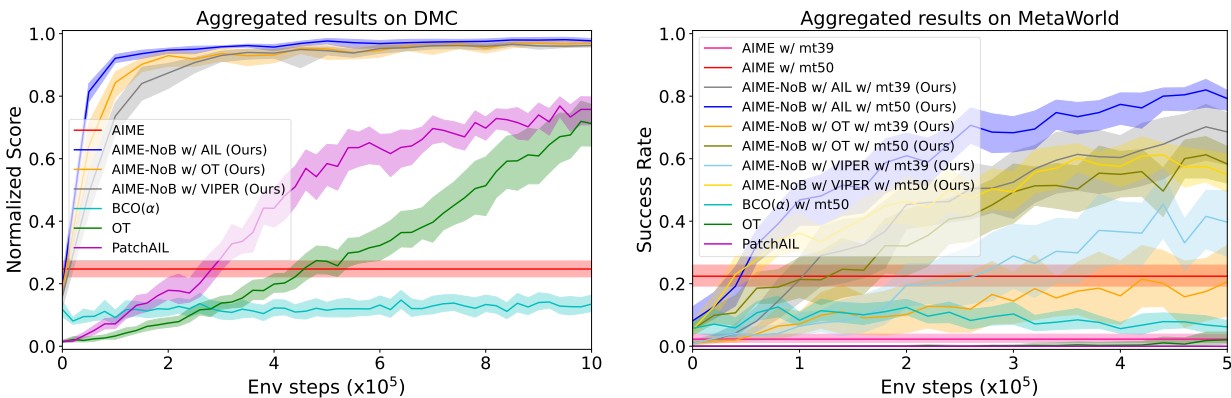

Figure 9: Comparing different variants of AIME-NoB with other algorithms. The figures show aggregated IQM scores on 9 DMC tasks and 6 MetaWorld tasks. All the algorithms are evaluated with 5 seeds with the shaded region representing 95% CI.

Figure 10 and Figure 11 respectively for DMC and MetaWorld. We also summarised the final performance after the interaction budget at Table 2 and Table 3 respectively for DMC and MetaWorld.

# I    Additional Experiments

**Do we need the whole embodiment datasets to establish the regulariser?** Although the data-driven regulariser is efficient, it requires to keep the entire embodiment datasets to build the regulariser. This can be challenging when the world model is pretrained on an internet-scale dataset. Therefore, we try to study if it is possible to use only a small portion of the dataset to establish the regulariser. We randomly sample a subset of the embodiment dataset as the regularisor and the result is shown in Figure 12. Surprisingly, the

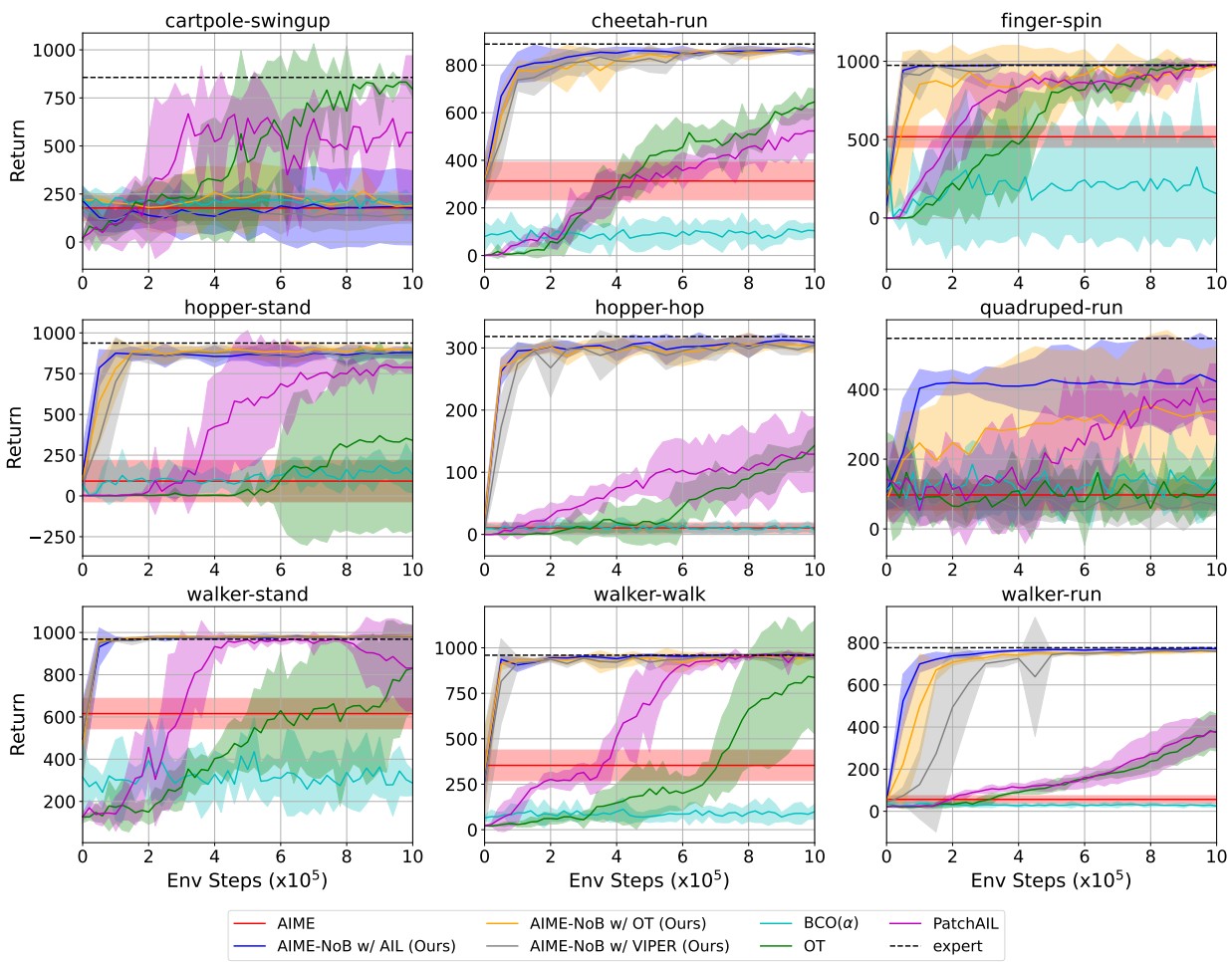

Figure 10: Benchmark results on 9 DMC tasks. Return are calculated by running the policy 10 times with the environment and taking the average return. The results are averaged across 5 seeds with the shaded region representing 95% CI.

Table 2: Final returns on DMC tasks at 1M environment steps. The best performance on each task is marked as **bold**.

|  | cartpole-swingup | cheetah-run | finger-spin | hopper-stand | hopper-hop | quadruped-run | walker-stand | walker-walk | walker-run |
|---|---|---|---|---|---|---|---|---|---|
| Expert | 856 | 888 | 974 | 937 | 318 | 546 | 968 | 959 | 776 |
| AIME | $177 \pm 64$ | $312 \pm 78$ | $518 \pm 67$ | $91 \pm 127$ | $10 \pm 8$ | $97 \pm 43$ | $616 \pm 72$ | $353 \pm 84$ | $56 \pm 17$ |
| BCO($\alpha$) | $206 \pm 30$ | $104 \pm 31$ | $154 \pm 258$ | $133 \pm 113$ | $7 \pm 6$ | $148 \pm 106$ | $286 \pm 58$ | $99 \pm 41$ | $25 \pm 5$ |
| OT | $\mathbf{795 \pm 40}$ | $645 \pm 54$ | $\mathbf{976 \pm 2}$ | $340 \pm 552$ | $143 \pm 29$ | $133 \pm 75$ | $830 \pm 200$ | $836 \pm 311$ | $373 \pm 88$ |
| PatchAIL | $569 \pm 400$ | $523 \pm 92$ | $\mathbf{979 \pm 5}$ | $787 \pm 36$ | $129 \pm 60$ | $371 \pm 101$ | $831 \pm 203$ | $\mathbf{962 \pm 5}$ | $378 \pm 75$ |
| AIME-NoB w/ AIL (Ours) | $177 \pm 192$ | $\mathbf{859 \pm 8}$ | $976 \pm 1$ | $878 \pm 31$ | $\mathbf{308 \pm 6}$ | $\mathbf{422 \pm 114}$ | $981 \pm 1$ | $954 \pm 5$ | $\mathbf{771 \pm 5}$ |
| AIME-NoB w/ OT (Ours) | $190 \pm 23$ | $\mathbf{856 \pm 8}$ | $968 \pm 13$ | $\mathbf{897 \pm 22}$ | $\mathbf{302 \pm 9}$ | $336 \pm 184$ | $\mathbf{983 \pm 3}$ | $952 \pm 4$ | $758 \pm 1$ |
| AIME-NoB w/ VIPER (Ours) | $141 \pm 42$ | $\mathbf{854 \pm 11}$ | $\mathbf{977 \pm 3}$ | $865 \pm 30$ | $\mathbf{304 \pm 4}$ | $68 \pm 29$ | $978 \pm 1$ | $\mathbf{945 \pm 12}$ | $759 \pm 4$ |

results show the action inference is even more stable when we only use 0.1% of the embodiment dataset, i.e. 2 trajectories, to establish the regulariser.

**Improving AIME-NoB on cartpole-swingup.** As shown in Figure 10, although AIME-NoB solves 8 out of 9 tasks, it still straggles at cartpole-swingup. We further investigate the cause of the low performance. After examination, we find issues in both of the datasets. For the demonstration dataset, we show the first

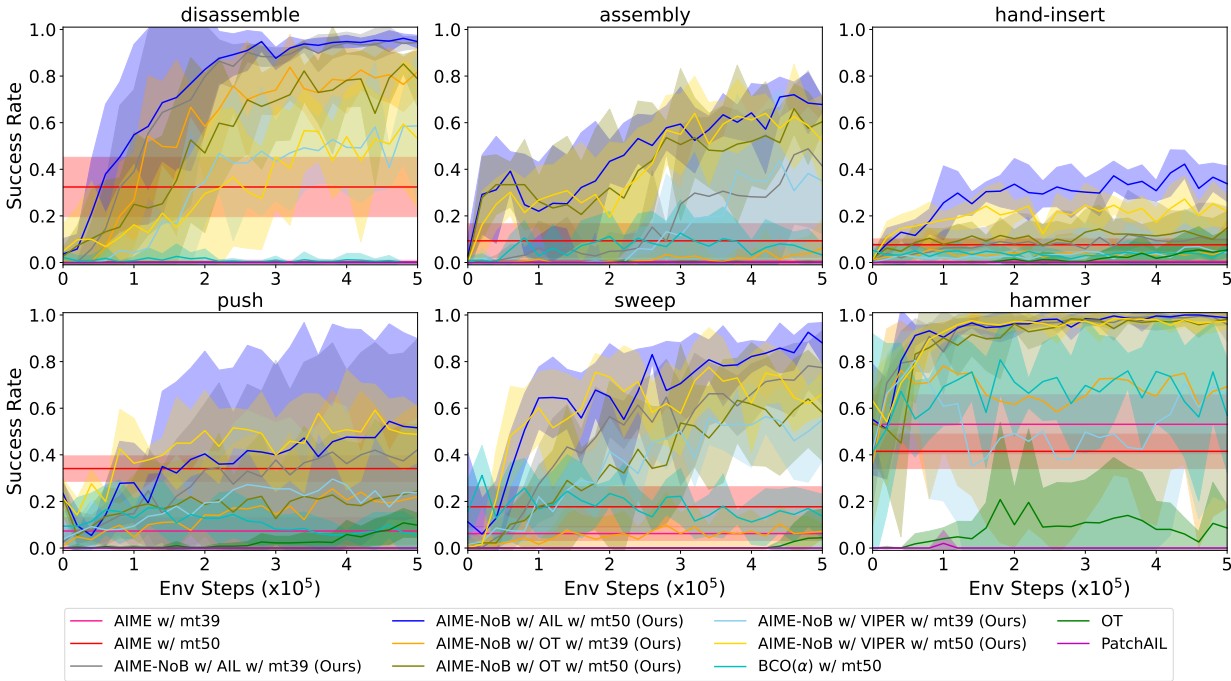

Figure 11: Benchmark results on 6 MetaWorld tasks. Trajectories are only counted as success when it success at the last time steps and the success rates are calculated with 100 policy rollouts. The results are averaged across 5 seeds with the shaded region representing 95% CI.

Table 3: Final success rate on MetaWorld tasks at 500K environment steps. The best performance on each task is marked as **bold**.

| | disassemble | assembly | hand-insert | push | sweep | hammer |
|---|---|---|---|---|---|---|
| AIME w/ mt39 | $0.00 \pm 0.01$ | $0.01 \pm 0.01$ | $0.00 \pm 0.01$ | $0.07 \pm 0.06$ | $0.06 \pm 0.03$ | $0.53 \pm 0.13$ |
| AIME w/ mt50 | $0.32 \pm 0.13$ | $0.09 \pm 0.07$ | $0.08 \pm 0.03$ | $0.34 \pm 0.05$ | $0.18 \pm 0.09$ | $0.42 \pm 0.07$ |
| BCO($\alpha$) w/ mt50 | $0.00 \pm 0.00$ | $0.03 \pm 0.05$ | $0.05 \pm 0.03$ | $0.06 \pm 0.06$ | $0.14 \pm 0.09$ | $0.57 \pm 0.25$ |
| OT | $0.00 \pm 0.00$ | $0.00 \pm 0.00$ | $0.05 \pm 0.08$ | $0.10 \pm 0.05$ | $0.04 \pm 0.12$ | $0.09 \pm 0.10$ |
| PatchAIL | $0.00 \pm 0.00$ | $0.00 \pm 0.00$ | $0.00 \pm 0.00$ | $0.00 \pm 0.00$ | $0.00 \pm 0.00$ | $0.00 \pm 0.00$ |
| AIME-NoB w/ AIL w/ mt39 (Ours) | $\mathbf{0.94 \pm 0.03}$ | $0.41 \pm 0.29$ | $0.10 \pm 0.14$ | $0.42 \pm 0.48$ | $0.77 \pm 0.16$ | $\mathbf{0.96 \pm 0.07}$ |
| AIME-NoB w/ AIL w/ mt50 (Ours) | $\mathbf{0.95 \pm 0.03}$ | $\mathbf{0.68 \pm 0.03}$ | $\mathbf{0.34 \pm 0.08}$ | $\mathbf{0.52 \pm 0.38}$ | $\mathbf{0.88 \pm 0.09}$ | $\mathbf{0.99 \pm 0.02}$ |
| AIME-NoB w/ OT w/ mt39 (Ours) | $0.82 \pm 0.11$ | $0.06 \pm 0.12$ | $0.05 \pm 0.13$ | $0.22 \pm 0.39$ | $0.07 \pm 0.03$ | $0.69 \pm 0.52$ |
| AIME-NoB w/ OT w/ mt50 (Ours) | $0.79 \pm 0.12$ | $0.60 \pm 0.04$ | $0.15 \pm 0.07$ | $0.24 \pm 0.29$ | $0.58 \pm 0.21$ | $\mathbf{0.98 \pm 0.02}$ |
| AIME-NoB w/ VIPER w/ mt39 (Ours) | $0.59 \pm 0.32$ | $0.35 \pm 0.25$ | $0.04 \pm 0.08$ | $0.24 \pm 0.32$ | $0.55 \pm 0.18$ | $0.66 \pm 0.47$ |
| AIME-NoB w/ VIPER w/ mt50 (Ours) | $0.53 \pm 0.30$ | $0.52 \pm 0.20$ | $0.22 \pm 0.07$ | $0.49 \pm 0.12$ | $0.66 \pm 0.09$ | $\mathbf{0.97 \pm 0.03}$ |

100 frames from 1 demonstration in Figure 13. As we can see, the initial position of the cart is from the center of the image with the pole pointing downward. The expert demonstration directly drive the cart to the left and then back to the right to swingup the pole and in the end balance the pole in the middle. Due to the carmera setup in cartpole, when the cart continuously to the left, the cart can move out of the scene, which makes the most important swing up process happen outside of the scene. This pose a severe challenge to the action inference and results in poor performance of AIME-NoB. We observe that all the policies from AIME-NoB learned to move left to go outside the scene in the beginning but comming back with an angle either not enough to push the pole to the up-right position or too much that the pole swing down from the right. For the embodiment dataset, it mainly contains data with a high speed rotating behaviour of the pole. Learning with imagination from these states will not lead to a helpful signal for swingup.

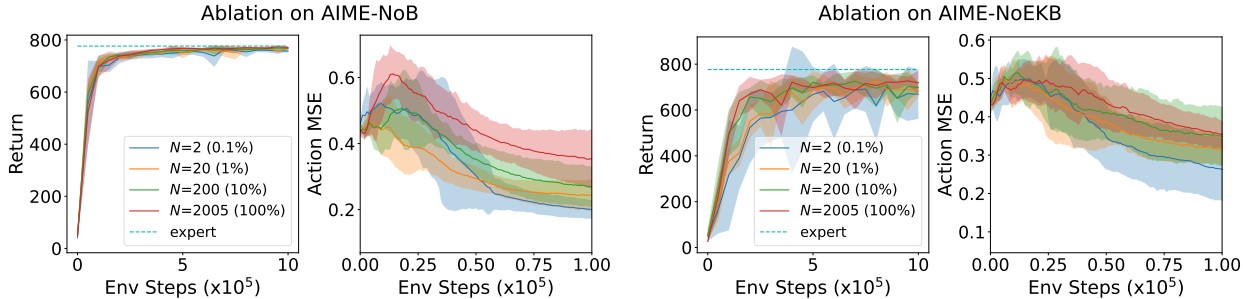

Figure 12: Ablations of the size of the data-driven regulariser $N$ on walker-run task. AIME-NoB is running with 10 demonstrations, while AIME-NoEKB is with 100 demonstrations. Action MSE is only shown for the first $10^5$ env steps. All results are averaged across 5 seeds with the shaded region representing a 95% CI.

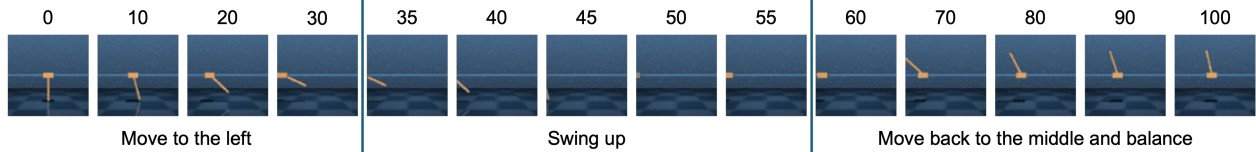

Figure 13: The first 100 frames of a cartpole swingup demonstration. The most important swingup behaviour is happened outside the scene.

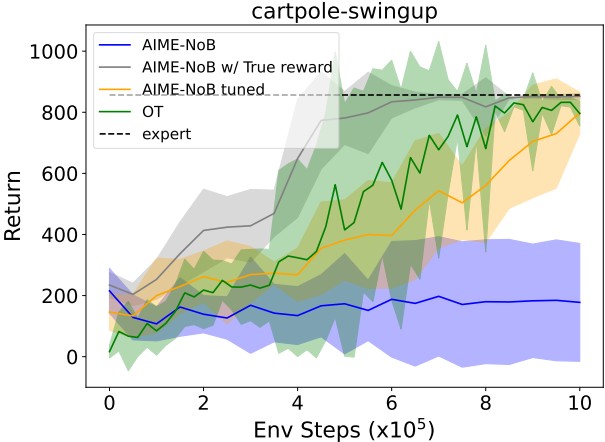

Figure 14: Results on cartpole-swingup with additional variants of AIME-NoB with the true reward and AIME-NoB with tuned hyper-parameters. Return are calculated by running the policy 10 times with the environment and taking the average return. The results are averaged across 5 seeds with the shaded region representing 95% CI.

In order to improve the performance, we need to have better emphasis on the role of the surrogate rewards. We remove the data-driven regulariser, i.e. setting $\alpha = 0$, since we know AIME loss won't help too much on this case and we want better utilisation of the online data. We further weaken the effect of AIME loss with a weight of 0.01. In Figure 14, we show the tuned version of AIME-NoB improve over the default hyper-parameters and can solve cartpole matching the best OT baseline. But still both of them have a large variance during training. In order to further understand the upper bound of this task, we also include a

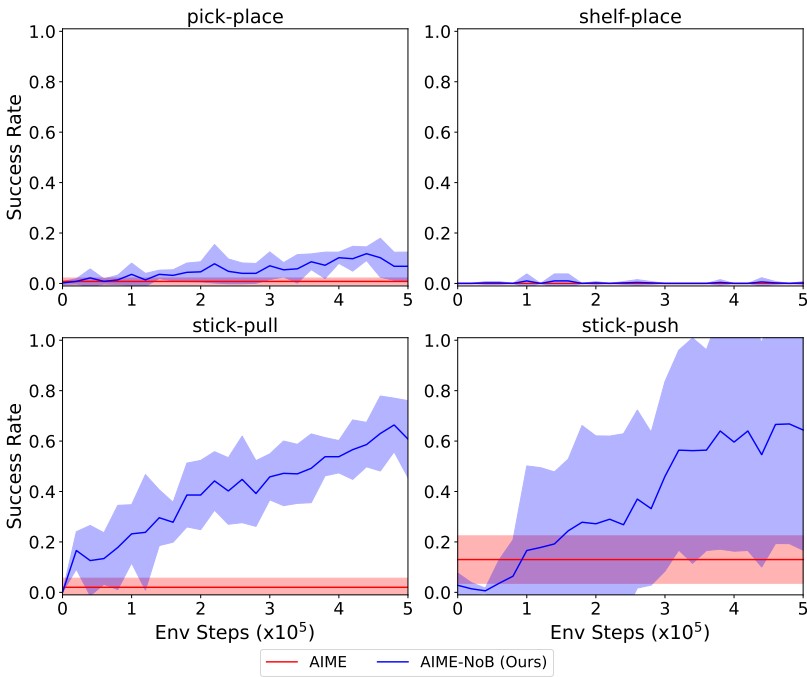

Figure 15: Benchmark results on additional 4 hard and very hard MetaWorld tasks. Trajectories are only counted as success when it success at the last time steps and the success rates are calculated with 100 policy rollouts. The results are averaged across 5 seeds with the shaded region representing 95% CI.

variant of AIME-NoB with true reward from the environment. We can see that with the true reward the task can be solved reliably. This motivates a better design of surrogate rewards in future works.

**More challenging tasks.** We extends our benchmarks with more challenging tasks. As from previous results in Figure 10 and Figure 11, baselines BCO($\alpha$), OT and PatchAIL are not well-performed, so we are not expecting them to be good on these even harder tasks. Thus, we mainly compare between AIME-NoB with AIME.

For MetaWorld, we include four additional tasks, namely pick-place, shelf-place, stick-pull and stick-push. According to Seo et al. (2022a), pick-place is a hard task and the other three are very hard tasks. The results are shown in Figure 15. From the results, AIME-NoB reliably outperforms AIME. Even in the very hard tasks stick-pull and stick-push, AIME-NoB manages around 60% success rates. However, AIME-NoB doesn't performs so well on pick-place and shelf-place. We conjecture it is due to the visual difficulties of the tasks. It is known that the world models based on reconstruction loss struggle at modeling small objects, which is the small cube we need to pick-up in these two tasks. Improving the world models' ability of modeling small objects or increase the resolutions of the observations will likely improve the performance.

For DMC, we conduct additional experiments on the humanoid embodiment. Since the humanoid embodiment is not in the initial list of the study, we essentially need to recollect the datasets on the embodiment and pretrain the world model. For the embodiment dataset, we use the same setting in the main benchmark to run Plan2Explore for 2M environment steps and use the 2000 trajectories in the replay buffer as the embodiment dataset. The world model is pretrained on the embodiment dataset for 200k gradient steps. For the demonstration dataset, we use the state-based policy from TD-MPC2 to collect 100 trajectories, and render the images to fit to our vision-based setting. Since humanoid is a very complex task, especially when using images as the observations, existing works (Yarats et al., 2021; Hafner et al., 2020) run the tasks for at least 30M environment steps. Due to the heavy computation load, we only run AIME-NoB for 10M environment steps. We show the results in Figure 16. We can clearly see AIME-NoB outperforms AIME and

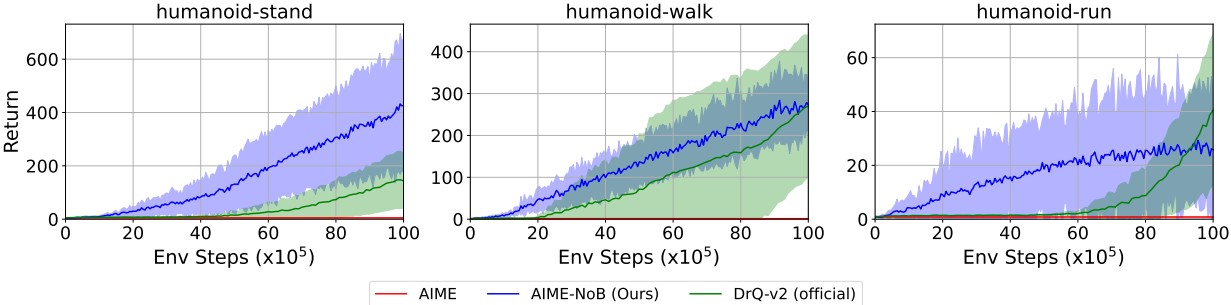

Figure 16: Results of AIME-NoB on 3 humanoid tasks. Returns are calculated by running the policy 10 times with the environment and taking the average return. AIME and AIME-NoB are averaged over 5 seeds; DrQ-v2 is over 10 seeds (Results from official repo `https://github.com/facebookresearch/drqv2/tree/main/curves`). The shaded region representing 95% CI.

it has the potential to further improve if training for more environment steps. Since there are no existing vision-based ILfO algorithms have been tested on humanoid, we add an additional comparison with the official results in DrQ-v2 which is a state-of-the-art vision-based model-free RL algorithm. We can see AIME-NoB is more sample-efficient than DrQ-v2 on all three tasks, but was caught up by DrQ-v2 around 10M steps on humanoid-walk and humanoid-run. We hypothesise that it is due to the limit capacity of the policy and value networks, which is implemented with 2-layers MLP with 128 units, cannot handle thus complex tasks. To the best of our knowledge, this is the first time that an ILfO algorithm shows progress on vision-based humanoid tasks.

**Comparing with Dreamer.** We further compare AIME-NoB with Dreamer which has access to the true reward provided by the environments. We compare with two versions, one is training Dreamer from scratch, one is training Dreamer but initialise the world model with the pretrained one, which we denote Dreamer w/ pt. One thing worth noting is although the pretrained world model has most of the components required for a new task, the reward decoder still needs to be trained from scratch since it is task-specific.

For DMC, we report the results at Figure 17. As we can see from the results, AIME-NoB achieves better sample efficiency on 8 out of 9 tasks except the problematic environment cartpole-swingup as discussed in Appendix I. On hopper-hop and quadruped-run, AIME-NoB is surpassed by Dreamer in the end. This is because as an imitation learning algorithm, AIME-NoB's performance is limited by the quality of the expert.

For MetaWorld, we report the results at Figure 18. As the manipulation tasks typically impose challenges for exploration, we see that Dreamer with or without the help of the pretrained model struggles to accomplish the task within the 500k environment steps. In the same time, with the help of the demonstrations, it is much easier for AIME-NoB to explore the related regions in the observation which results in a better sample efficient. This marks an advantage of using demonstrations over using a scalar reward to define the task.

**Performance gain of AIME-NoB regarding the same computation time.** Arguably, due to the additional online interactions with the environment, fintuning the model and training the actor-critic, on our implementation AIME-NoB needs approximately ×2.86 times per gradient step comparing with the base algorithm AIME. In this perspective, it is reasonable to ask how compute efficient is AIME-NoB comparing with AIME. We redraw the benchmark results from Figure 2 with respect to the compute time in Figure 19. From the results, we can conclude that, if the online interaction is allowed, we should always use AIME-NoB over AIME even when the compute budget is tight.

**Presence of EKB and DKB on more tasks.** In the main text, we only show the presence of EKB and DKB on walker-run with Figure 1 and Figure 3. Here we show more evidence on hopper-hop from the DMC suite and the disassemble task from the MetaWorld suite in Figure 20. As we can see, different tasks can have different distributions of the two barriers. For hopper-hop, AIME and the MBBC oracle almost overlap with each other, meaning the EKB is negligible in this case, and the performance gap is mainly attributed to DKB.

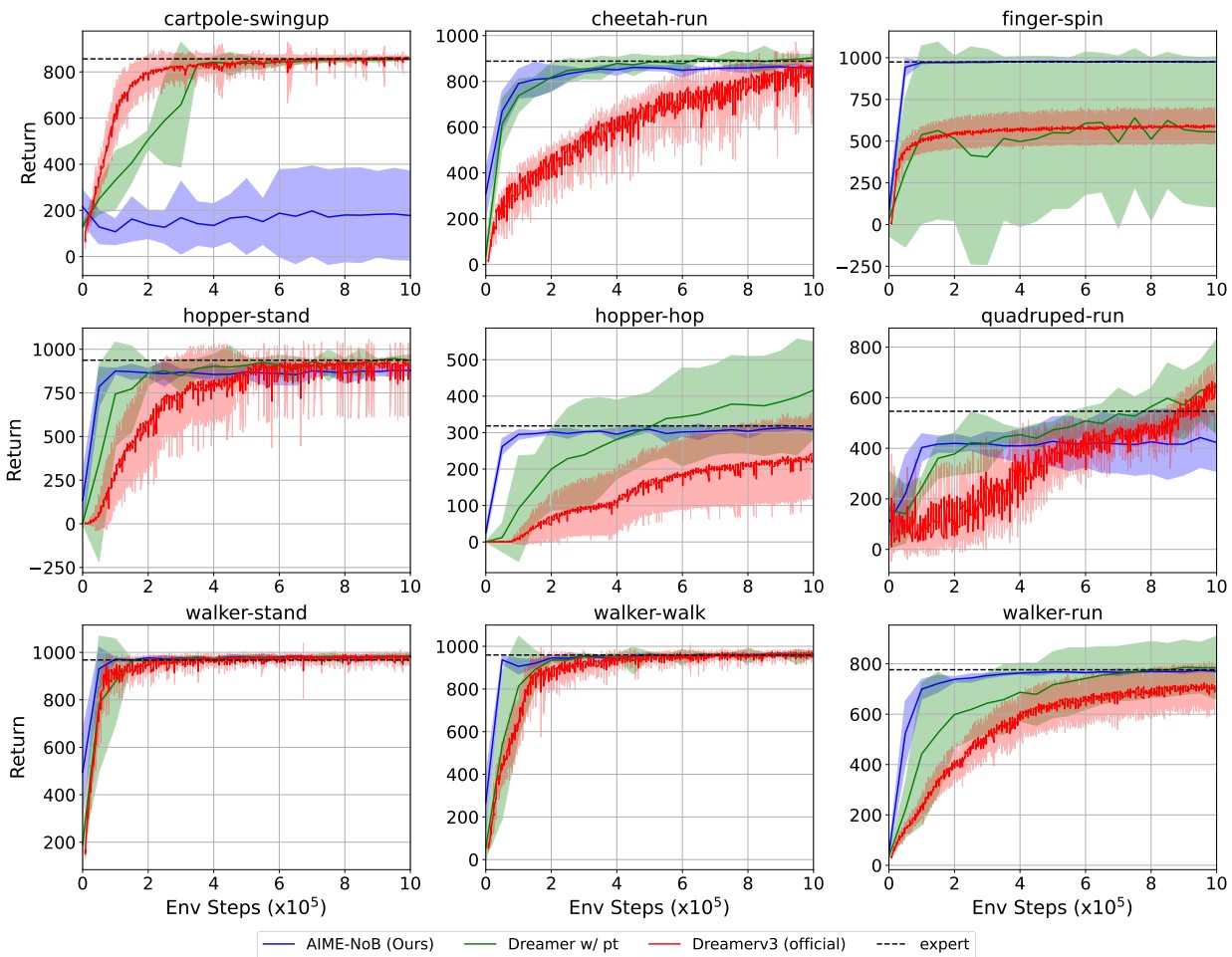

Figure 17: Additional comparison between AIME-NoB and Dreamer on 9 DMC tasks. Returns are calculated by running the policy 10 times with the environment and taking the average return. The results are averaged across 5 seeds (10 seeds for Dreamerv3 (official) from official repo `https://github.com/danijar/dreamerv3/tree/main/scores`) with the shaded region representing 95% CI.

This is because the Plan2Explore embodiment dataset already contains hopping behavior, and the pretrained model has enough knowledge to infer the correct actions. For the disassemble task from MetaWorld, the trend is more similar to Walker-Run, where both barriers are present. When the number of demonstrations is low, DKB is dominant, and as the number of demonstrations increases, EKB and DKB exhibit more equal strength.

## J   Upper bounds for EKB and DKB

In the section, we propose upper bounds for both of the barriers based on our intuition. Without losing any generality, we assume that the reward at each step is bounded by a scalar $r_{\max}$, i.e. $|r_t| \leq r_{\max}$.

**Upper bound for EKB** Our analysis is inspired by Xu et al. (2020). We assume the policy $\pi$ is expressive enough to fit the learning objective, then we have $\pi_{\omega^*} \equiv \hat{p}_{\pi_{\text{demo}}}$ and $\pi_{\psi^*} \equiv q_{\phi^*}$. Thus, we have

$$D_{\text{KL}}(\pi_{\omega^*}(a_t|o_{1:t}) \,|\, \pi_{\psi^*}(a_t|o_{1:t})) = D_{\text{KL}}(\hat{p}_{\pi_{\text{demo}}}(a_t|o_{1:T}) \,|\, q_{\phi^*}(a_t|o_{1:T})). \tag{17}$$

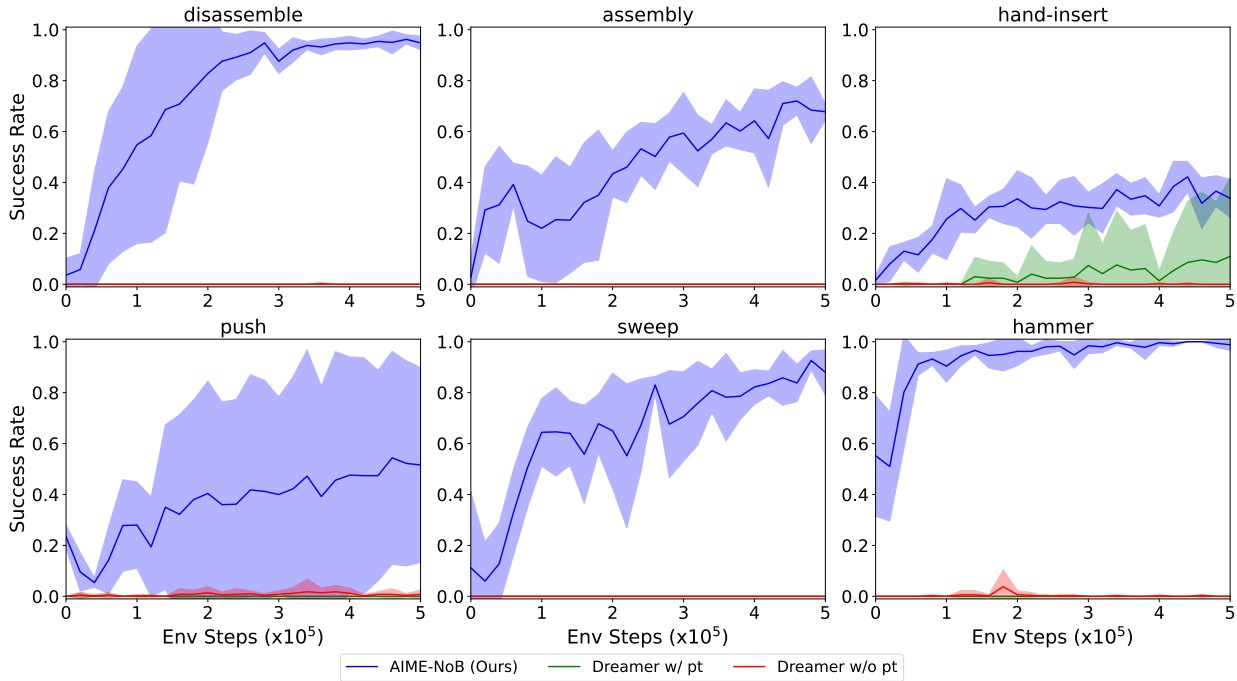

Figure 18: Additional comparison between AIME-NoB and Dreamer on 6 MetaWorld tasks. Trajectories are only counted as success when it is marked successful at the last time steps and the success rates are calculated with 100 policy rollouts. The results are averaged across 5 seeds with the shaded region representing 95% CI.

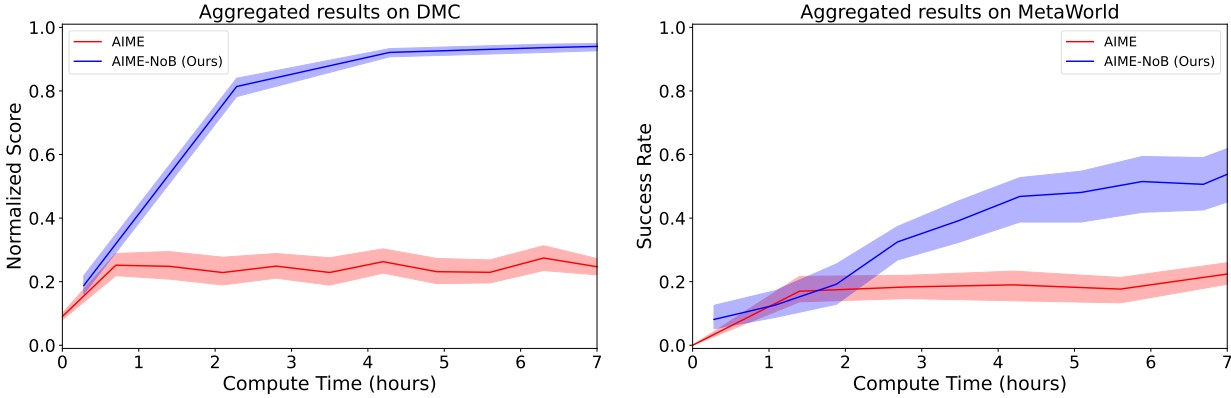

Figure 19: Comparing AIME-NoB with AIME on the same compute budget. The figures show aggregated IQM scores on 9 DMC tasks and 6 MetaWorld tasks. All the algorithms are evaluated with 5 seeds on each task and the shaded region representing 95% CI. AIME-NoB is not starting from 0 since the first point is evaluate after the 2000 pretraining steps with AIME.

We assume the inference error is bounded by $\epsilon$, i.e. $\mathbb{E}_{o_{1:T} \sim D_{\mathrm{demo}}}[D_{\mathrm{KL}}(\hat{p}_{\pi_{\mathrm{demo}}}(a_t|o_{1:T}) \,|\, q_{\phi^*}(a_t|o_{1:T}))] \leq \epsilon$. Then, according to Theorem 1 in Xu et al. (2020), we have the upper bound of EKB as

$$\mathrm{EKB} = \mathcal{R}(\pi_{\omega^*}) - \mathcal{R}(\pi_{\psi^*}) \leq 2\sqrt{2}r_{\max}T(T+1)\epsilon. \tag{18}$$

The term $1/(1-\gamma)^2$ from the original theorem is replaced by $T(T+1)$ as we are considering a finite horizon problem in this paper. The proof can be done by redoing their proof on the finite horizon case without the discount factor $\gamma$, i.e. for case where $1-\gamma$ serves as the normalizing factor we have $1-\gamma = T^{-1}$ while for

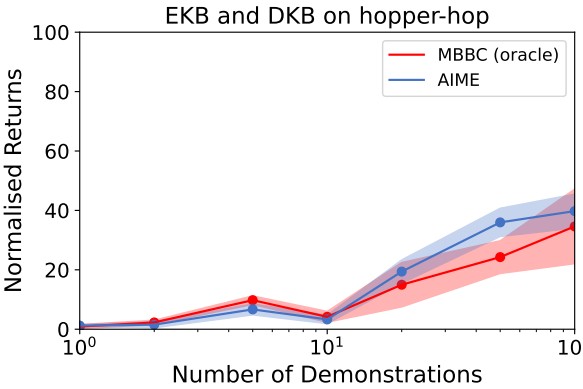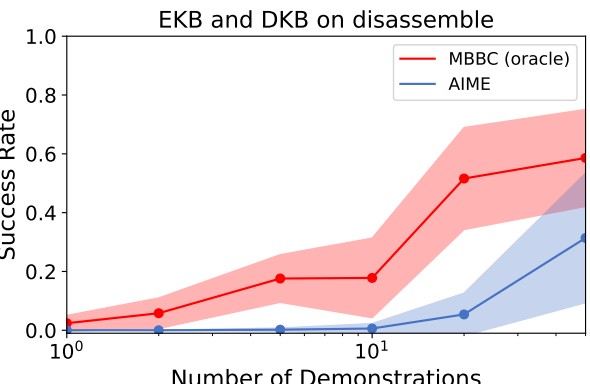

Figure 20: Additional evidence of the presence of EKB and DKB on hopper-hop from DMC and disassemble from MetaWorld. The results are averaged across 5 seeds with the shaded region representing 95% CI.

case where it is used as a discount factor we have $\gamma = 1$. The bound is tightened when the inference error $\epsilon$ decreases and EKB is completely overcome when we reduce the inference error to 0.

**Upper bound for DKB** Without losing generality, we assume the initial state of each trajectory is sampled from a discrete distribution $\rho(s)$. Then, the expected return can be rewritten as $\mathcal{R}(\pi) = \sum_s \rho(s) \mathcal{R}(\pi)_{|s_0=s}$. Based on this definition, we can rewrite DKB with a factorization based on the support of the demonstration dataset, i.e.

$$
\begin{aligned}
\text{DKB} = \mathcal{R}(\pi_{\text{demo}}) - \mathcal{R}(\pi_{\omega^*}) = \sum_{s \in D_{\text{demo}}} & \rho(s)(\mathcal{R}(\pi_{\text{demo}})_{|s_0=s} - \mathcal{R}(\pi_{\omega^*})_{|s_0=s}) \\
+ \sum_{s \notin D_{\text{demo}}} & \rho(s)(\mathcal{R}(\pi_{\text{demo}})_{|s_0=s} - \mathcal{R}(\pi_{\omega^*})_{|s_0=s})
\end{aligned}
\tag{19}
$$

Given the policy expressiveness assumption, the learned policy $\pi_{\omega^*}$ can match the performance of $\pi_{\text{demo}}$ on the support of the demonstrations. Therefore, the first term in Equation (19) becomes 0. Thus, DKB only depends on the performance on the initial states out of support of the demonstration dataset. Without any assumption on the generalization of the learned policy, it can perform arbitrarily bad on the OOD states and in the worst case we have $\mathcal{R}(\pi_{\omega^*}) = -Tr_{\max}$. But in practice, we assume the worst performance of the learned policy can be bounded by $R_{\min}^{\pi_{\omega^*}}$, i.e. for all $s \notin D_{\text{demo}}$ we have $\mathcal{R}(\pi_{\omega^*})_{|s_0=s} \geq R_{\min}^{\pi_{\omega^*}}$. We denote the support ratio of the demonstrations as $\eta = \sum_{s \in D_{\text{demo}}} \rho(s) / \sum_s \rho(s) = \sum_{s \in D_{\text{demo}}} \rho(s)$, then we have the upper bound for DKB as

$$
\text{DKB} \leq (1-\eta)(Tr_{\max} - R_{\min}^{\pi_{\omega^*}}) \leq 2(1-\eta)Tr_{\max}.
\tag{20}
$$

It can be clearly seen from the upper bound that when we increase the support ratio $\eta$, the bound is tightened. If increasing support ratio is not possible as the demonstrations are expensive, we can also increase the performance on the OOD states, i.e. $R_{\min}^{\pi_{\omega^*}}$, by providing extra learning signals.

