# OpenReview forum: "Overcoming Knowledge Barriers: Online Imitation Learning from Visual Observation with Pretrained World Models"
_TMLR — Accepted by TMLR_

### Review · Reviewer_N6FK · 2025-02-12

**Summary Of Contributions:**

The paper targets two critical issues in imitation learning from observation (ILfO) when leveraging pretrained world models—namely, the Embodiment Knowledge Barrier (EKB) and the Demonstration Knowledge Barrier (DKB). In response to these challenges, the authors propose AIME-NoB, an extension of the earlier AIME algorithm. The key contributions include:

- EKB Mitigation: The paper proposes an online interaction phase combined with a data-driven regularizer to continuously update the world model. The rationale is that by collecting new data online, the pretrained model can better handle out-of-distribution observations and actions that were not encountered during pretraining.
- DKB Mitigation: To address the limited coverage of expert demonstrations, the authors introduce a surrogate reward function derived from a VIPER-style video prediction model. This reward is then used in a dreamer-style actor-critic framework to extend the policy’s guidance beyond the sparse demonstration data.
- Experiments are carried out on DMControl and MetaWorld benchmarks. Detailed ablation studies are presented to analyze the effect of the regularizer ratio (α) and the value gradient weight (β) on performance, with additional discussion of challenging tasks such as cartpole swingup and quadruped-run.

**Audience:**

Yes

**Broader Impact Concerns:**

No Broader Impact Concern required.

**Claims And Evidence:**

Yes

**Requested Changes:**

1. The authors should provide clear, formal definitions of both the Embodiment Knowledge Barrier and the Demonstration Knowledge Barrier. This could include mathematical formulations or metrics that quantify the gap between the oracle and learned models. A rigorous treatment would help readers better appreciate the extent to which each barrier is mitigated by the proposed modifications.

2. It is important for the authors to more clearly differentiate their approach from prior work. For the online interaction and surrogate rewards, the paper should emphasize what is non-trivial about this integration.

3. Figure 1 and the accompanying discussion should more explicitly show how the introduction of online interactions, coupled with the data-driven regularizer, reduces the EKB. If possible, providing quantitative or visual evidence would help substantiate the claim.

4. The paper would benefit from experiments on more challenging tasks or from comparisons with alternative methods that use hand-designed or learned surrogate rewards. Furthermore, a more detailed discussion on how the proposed method could be adapted or scaled to address real-world robotics problems—where domain shifts and action space mismatches are significant—would enhance the practical impact of the work.

**Strengths And Weaknesses:**

**Strengths:**

- The paper draws attention to the overlooked issues of knowledge barriers in ILfO. The distinction between EKB (the inability of pretrained models to generalize to unseen embodiments) and DKB (the shortcomings stemming from limited demonstrations) is intuitively appealing and addresses an important gap in current methods.

- The study is backed by extensive experiments on two benchmark suites. The authors provide ablation studies that reveal how varying key hyperparameters ($\alpha$ and $\beta$) impact performance. This empirical analysis lends support to the claim that their modifications yield improvements in both sample efficiency and final performance.

**Weaknesses:**

- The definitions of the Embodiment Knowledge Barrier (EKB) and Demonstration Knowledge Barrier (DKB) remain largely informal. While the authors offer intuitive descriptions, the lack of a rigorous, formal definition—and quantitative metrics that measure these barriers—limits the clarity of the paper. As a result, it is difficult to assess precisely how much each modification reduces the corresponding gap.

- The methodological contribution appears largely incremental. The proposed solutions for both barriers are essentially straightforward extensions of known techniques by including more information. For EKB, the solution is to allow online interactions with a data-driven regularizer—a sensible but not conceptually novel modification. This is compounded by the observation that Figure 1 does not clearly illustrate how these additions overcome the EKB beyond merely providing additional “oracle” information.

- The contributions of the paper are more in the realm of effective engineering, i.e., combining online fine-tuning with surrogate rewards. While the empirical improvements are significant in some benchmarks, the overall methodological novelty remains modest, and the practical implications for real-world systems (e.g., robotics) are somewhat speculative.

---

> ### Author Response · Authors · 2025-02-26
> **Rebuttal 1/2**
>
> Thanks for your time to review our paper. We would like to address your questions and requests below:
>
> > The authors should provide clear, formal definitions of both the Embodiment Knowledge Barrier and the Demonstration Knowledge Barrier. This could include mathematical formulations or metrics that quantify the gap between the oracle and learned models. A rigorous treatment would help readers better appreciate the extent to which each barrier is mitigated by the proposed modifications.
>
> We would like to clarify that formal definitions of EKB and DKB are already provided in **Section 3 (Equations 4 and 5)**, where we present a quantitative measure of these barriers. Additionally, **Sections 3.1 and 3.2** explain how these formulations motivate our proposed solutions. We encourage the reviewer to revisit these sections, as they directly address this request.
>
>  > Figure 1 and the accompanying discussion should more explicitly show how the introduction of online interactions, coupled with the data-driven regularizer, reduces the EKB. If possible, providing quantitative or visual evidence would help substantiate the claim.
>
> We have revised **Figure 1 and its caption** to clarify that EKB and DKB refer to the **value difference at evaluation points in the same column (same number of demonstrations), rather than the integral of the region**.
>
> Additionally, we would like to emphasize that **Figure 1 serves as motivation for the paper, not a depiction of our solution**. The effects of our approach are analyzed in **Section 4.3 (Q2) and Figure 3**, where we show that **AIME-NoEKB (our method without the surrogate reward) successfully mitigates EKB, and AIME-NoB (full method) further resolves DKB**.
>
> > It is important for the authors to more clearly differentiate their approach from prior work. For the online interaction and surrogate rewards, the paper should emphasize what is non-trivial about this integration.
>
> We would like to clarify that our study is motivated by specific challenges in Imitation Learning from Observation (ILfO) when using pretrained models. We introduce a novel framework that explicitly decomposes the imitation gap into two components: the Embodiment Knowledge Barrier (EKB) and the Demonstration Knowledge Barrier (DKB). Based on this theoretical foundation, we propose targeted solutions—online interaction with a regularizer to mitigate EKB and a surrogate reward function to address DKB. These are implemented on top of the state-of-the-art AIME algorithm, resulting in our method, AIME-NoB.
>
> We believe our paper provides valuable insight for the ILfO field and broaden the applicability of the pretrained world model. These are the unique contribution of our paper comparing with existing works.
>
> > The paper would benefit from experiments on more challenging tasks or from comparisons with alternative methods that use hand-designed or learned surrogate rewards. Furthermore, a more detailed discussion on how the proposed method could be adapted or scaled to address real-world robotics problems—where domain shifts and action space mismatches are significant—would enhance the practical impact of the work.
>
> We would like to kindly point out we have already provided these experiments in the Appendix I. To be specific:
> - For more challenging tasks, we have added Figure 15 to test on the leftover four very-hard tasks from the MetaWorld benchmark and Figure 16 to test on three changeling vision-based humanoid tasks. AIME-NoB still lead the performance on these challenging tasks.
> - For comparisons with hand-design reward, we have added Figure 16, Figure 17 and Figure 18 to compare with online RL methods, i.e. DrQ-v2 and Dreamerv3, that use the ground truth reward from the simulator. Besides the online RL methods, we also compare with a version to use our pretrained world model with the ground truth reward. All the comparison showcase the strong performance of AIME-NoB.
>
> We appreciate the reviewer’s interest in these analyses and encourage them to review **Appendix I for further details**.

---

> ### Author Response · Authors · 2025-02-26
> **rebuttal 2/2**
>
> Besides these responses to the questions and the requirement, we would also like to clarify some misunderstandings:
>
> > To address the limited coverage of expert demonstrations, the authors introduce a surrogate reward function derived from a VIPER-style video prediction model. This reward is then used in a dreamer-style actor-critic framework to extend the policy’s guidance beyond the sparse demonstration data.
>
> We have to point out that this statement is **not an accurate reflection of our method**. The core equation for the DKB solution is eq. 9 where we combine the Imitation loss from AIME (first term) and a dreamer-style value gradient loss (second term). The second term, together with eq. 7 and eq. 8, is a common practice since Dreamer [1] and doesn't have any tie with VIPER. VIPER only comes into play when we consider different possibilities to define the surrogate reward $r^{\mathrm{sur}}$, and VIPER play as one of the three possibilities together with OT and AIL. All these three variant acts independently, i.e. when we use OT or AIL as the surrogate reward, VIPER won't be used. In the ablation study (Q3.1 in page 9 and Figure 4) shows AIL variant performs the best and VIPER variant performs the worst. That's why we are AIL as the default choice of the surrogate reward, not VIPER.
>
> Thus, while VIPER appears in our paper **as a comparative baseline**, it is **not central to our approach**. Referring to our method as **"derived from a VIPER-style video prediction model"** creates **a misleading impression** of our contribution. We respectfully ask the reviewer to reconsider this characterization.
>
> [1] Danijar Hafner, Timothy Lillicrap, Jimmy Ba, and Mohammad Norouzi. Dream to Control: Learning Behaviors by Latent Imagination. In ICLR 2020

---

> > ### Comment · Reviewer_N6FK · 2025-03-30
> >
> > Thank you for your detailed feedback. I encourage the authors to incorporate the points raised in the rebuttal into the revised manuscript. While I continue to view the reward function as utilizing a VIPER-style loss, I do not believe this aspect substantially detracts from the overall contributions of the work.

---

### Review · Reviewer_Cwx7 · 2025-02-20

**Summary Of Contributions:**

This paper addresses fundamental challenges in Imitation Learning from Observation. It identifies two primary barriers: the Embodiment Knowledge Barrier (EKB) and the Demonstration Knowledge Barrier (DKB). The EKB is caused when pretrained models encounter novel observations and struggle to infer correct actions, while the DKB results from the limited generalization of policies trained on sparse expert demonstrations. The authors propose AIME-NoB, an extension of the AIME algorithm, which uses online interactions and a data-driven regularizer to mitigate the EKB and a surrogate reward function to expand state-space coverage and address the DKB. The paper provides a thorough empirical evaluation, demonstrating significant improvements in sample efficiency and final performance on the DeepMind Control Suite and MetaWorld benchmarks.

**Audience:**

Yes

**Broader Impact Concerns:**

None.

**Claims And Evidence:**

Yes

**Requested Changes:**

Surrogate rewards: It would be nice to provide more justification or explanation for why AIL seems to perform better than OT or VIPER. Are there possible theoretical reasons for this? Is the benefit of AIL particular to this problem setting or do you think it more broadly superior as a surrogate reward?

Discuss computational overhead: Since AIME-NoB incorporates online interactions, surrogate rewards, and regularization, a more explicit comparison of computational costs vs. performance gains would be valuable.

Consider alternative online training schedules: The current method balances pretraining data and online interactions via a static α parameter. Could an adaptive scheduling approach improve stability?

**Strengths And Weaknesses:**

Strengths
+ The introduction of EKB and DKB provides a well-motivated framework for analyzing limitations in ILfO.
+ The integration of online interactions and surrogate rewards effectively tackles the identified barriers.
+ The experimental results span many tasks across two benchmarks, demonstrating superior performance compared to baselines.
+ Various ablations offer insight into the contributions of different components of AIME-NoB.

Weaknesses
- Limited Theoretical Justification: While the empirical findings are strong, a more rigorous theoretical analysis of the knowledge barriers and proposed solutions would enhance the work, but may be out of scope.
- Complexity of Implementation: AIME-NoB involves multiple interdependent components (regularization, surrogate reward, online interaction), which may make it harder to reproduce. It would be great if the authors can open source all their code and experimental harnesses so others can more easily build on this work.

---

> ### Author Response · Authors · 2025-02-26
>
> We would like to thank the reviewer for the valuable feedback.  Please kindly find our response to your concerns below:
>
> > Limited Theoretical Justification: While the empirical findings are strong, a more rigorous theoretical analysis of the knowledge barriers and proposed solutions would enhance the work, but may be out of scope.
>
> While we agree with you that our paper is light theoretical contribution, missing for example a proof of convergence or guarantee. But we think our formal definitions of the EKB and DKB is already enough for an empirical paper to clarify the problem and serves as a starting point for the following work to make in-depth theoretical analysis.
>
> > Complexity of Implementation: AIME-NoB involves multiple interdependent components (regularization, surrogate reward, online interaction), which may make it harder to reproduce. It would be great if the authors can open source all their code and experimental harnesses so others can more easily build on this work.
>
> Thanks for your advice. We would like to point out the source code is already available in the "Supplementary Material" for this submission. Upon the acceptance of this paper, we will release the code on GitHub alongside all the intermediate results, e.g. the models, datasets and the learning curves. We hope this can help the community to build on this work easily.
>
> > Surrogate rewards: It would be nice to provide more justification or explanation for why AIL seems to perform better than OT or VIPER. Are there possible theoretical reasons for this? Is the benefit of AIL particular to this problem setting or do you think it more broadly superior as a surrogate reward?
>
> Thanks for your question. We would like to bring your attention to Q3.1 in page 9, where we attribute the better performance of AIL variant to the direct adaptation of the discriminator during online training. We would like to elaborate on the level of adaptation of each of the three variants:
> - The VIPER variant has zero adaptation. The VIPER model is a separate model besides the world model, and it is pretrained on the demonstration dataset and frozen during online training.
> - The OT variant has an indirect adaptation. Although computing the OT reward is mostly parameter free, in the current design it is based on the image encoder from the world model. The image encoder of the world model will be fine-tuned during the online training as it sees more task-related observations, which provides the OT reward with an indirect adaptation to the task.
> - The AIL variant has a direct adaptation. During online training, the discriminator is trained together with the world model, so that it can dynamically adapt to the aspect that the agent should care about currently.
>
> We can see in Figure 4 (a) and (b), AIL performs the best among all with the highest level of adaptation. We have also shown in Figure 7 of Appendix G that after the online training, the AIL reward has a more linear correlation with the true reward.
>
> We conjecture that if we can enhance the VIPER or the OT variant with more adaptability, the performance of these variant may also increase. But it will demand new design of these variants and is beyond the scope of this paper. We have added this analysis to Q3.1 on page 9 to aid understanding.
>
> > Discuss computational overhead: Since AIME-NoB incorporates online interactions, surrogate rewards, and regularization, a more explicit comparison of computational costs vs. performance gains would be valuable.
>
> Thanks for your question. In the revised version, we add Figure 19 in Appendix I to compare AIME-NoB with AIME with the same computation budget of 7 hours. As the results show, AIME-NoB is more preferable even if the computation budget is tight, e.g. less than 3 hours.
>
> Regarding the different variants of AIME-NoB, although the AIL version introduces the additional discriminator, the contribution of the compute is negligible.
>
> > Consider alternative online training schedules: The current method balances pretraining data and online interactions via a static α parameter. Could an adaptive scheduling approach improve stability?
>
> Thanks for your question. We would like first bring your attention to Figure 5 where we ablate the influence of $\alpha$. We conclude from this figure that we can gain more stability by choosing a higher $\alpha$ but at a cost of sample efficiency. Furthermore, the **append** variant corresponds to an inverse proportional schedule of $\alpha$ from 1.0 to 0.66 during the course of training. As we can see, the Action MSE is very stable, but this variant is also the least sample efficient. Although we believe a better scheduling strategy could exist that balances between the stability and sample efficiency, it is largely an engineering problem which will add more hyperparameters. We think that is beyond the scope of this paper. We add this discussion in Q3.3 on page 10.

---

> > ### Comment · Reviewer_Cwx7 · 2025-03-26
> > **Good response**
> >
> > Thank you for your response. The authors have addressed my main concerns.

---

### Review · Reviewer_WXaM · 2025-02-24

**Summary Of Contributions:**

This paper presents an algorithm that improves existing methods for finetuning a pretrained policy on novel ILfO tasks. The paper first proposes to decomposes the performance gap between policy and oracle with 1. DKB, which measures the gap between oracle with methods having access to GT action, and 2. EKB, which measures the performance gap between the . The paper then proposes modifications to existing ILfO algorithms that are motivated to addressing such gaps ("knowledge barriers"). To address EKB, the paper proposes to finetune the pretrained world model on new interaction data. To address DKB, the paper proposes to use surrogate reward function and trains the policy in dreamer style.

**Audience:**

Yes

**Claims And Evidence:**

Yes

**Requested Changes:**

Same as the weaknesses section. Mainly, real-world robotics experiments, and analysis on questions raised above.

**Strengths And Weaknesses:**

Strengths
- The paper is well-written, and tackles an important problem which is finetuning pretrained policy on novel tasks.
- Extensive ablation study

Weaknesses
- It looks like technical wise the method is somewhat "finetuning a policy and world model with dreamer and surrogate reward", yet motivated in a relatively convoluted way.
- My main concern is that all experiments are run with simulation / toy examples. How well does the method work in real-world robotics tasks?
- For figure 1, how does the figure look with other tasks / domains? What makes a task having more DKB / EKB? Does the method and baseline perform differently with different observation space?

---

> ### Author Response · Authors · 2025-02-26
>
> We thank the reviewer for their valuable feedback. Below, we address the concerns raised.
>
> > It looks like technical wise the method is somewhat "finetuning a policy and world model with dreamer and surrogate reward", yet motivated in a relatively convoluted way.
>
> We would like to clarify that our study is motivated by specific challenges in Imitation Learning from Observation (ILfO) when using pretrained models. We introduce a novel framework that explicitly decomposes the imitation gap into two components: the Embodiment Knowledge Barrier (EKB) and the Demonstration Knowledge Barrier (DKB). Based on this theoretical foundation, we propose targeted solutions—online interaction with a regularizer to mitigate EKB and a surrogate reward function to address DKB. These are implemented on top of the state-of-the-art AIME algorithm, resulting in our method, AIME-NoB.
>
> While our approach does involve fine-tuning a policy and world model, the key contribution lies in our systematic analysis and the proposed solutions tailored for ILfO. Simply describing AIME-NoB as _"fine-tuning with Dreamer and a surrogate reward"_ overlooks the theoretical motivation and methodological contributions of our work.
>
> > My main concern is that all experiments are run with simulation / toy examples. How well does the method work in real-world robotics tasks?
>
> While real-world experiments are always valuable, our study already includes an extensive evaluation. In addition to the 15 tasks in the main text, we provide results on 7 additional challenging environments in the appendix, including vision-based humanoid locomotion tasks, which are known to be particularly difficult. Despite these challenges, AIME-NoB achieves strong performance, even outperforming a state-of-the-art RL method that has access to ground-truth rewards. All the envrionments and tasks we tested on are standard benchmarks in the field, some of them previously unsolved in the ILfO setting.
>
> Given the diversity and difficulty of our test environments, and to our knowledge the most extensive experimental evaluation in the field of ILfO, we believe our results sufficiently demonstrate the effectiveness of AIME-NoB and provide strong evidence for its applicability.
>
> > For figure 1, how does the figure look with other tasks / domains? What makes a task having more DKB / EKB? Does the method and baseline perform differently with different observation space?
>
> Thank you for the insightful questions.
>
> - **Other tasks/domains:** Additional plots similar to Figure 1 are provided in Appendix I Figure 20 for hopper-hop and disassemble. For hopper-hop, AIME and the MBBC oracle almost overlap with each other, meaning the EKB is negligible in this case, and the performance gap is mainly attributed to DKB. This is because the Plan2Explore embodiment dataset already contains hopping behaviour, and the pretrained model has enough knowledge to infer the correct actions. For the disassemble task from MetaWorld, the trend is more similar to walker-run, where both barriers are present. When the number of demonstrations is low, DKB is dominant, and as the number of demonstrations increases, EKB and DKB exhibit more equal strength.
> - **Task factors affecting EKB and DKB in general:**
>     - **EKB** is influenced primarily by the discrepancy between the embodiment dataset and the target task. For instance, if the dataset consists only of walking behaviors, the EKB will be smaller for a related task like running compared to an unrelated task like backflipping. Additionally, EKB increases with more demonstrations due to accumulated inference errors, as observed in Figure 1.
>     - **DKB** is mainly determined by the number of demonstrations—more demonstrations reduce DKB. It is also influenced by task diversity; for example, in manipulation tasks, a fixed object and goal position lead to lower DKB compared to randomly assigned positions.
> - **Effect of observation space:** Prior work (Figure 4 in [1]) suggests that all algorithms exhibit some variance in performance across observation spaces, yet both EKB and DKB persist across conditions. In our study, we focus on the most challenging case, where improvements have the highest impact.
>
> [1] Xingyuan Zhang, Philip Becker-Ehmck, Patrick van der Smagt, and Maximilian Karl. Action Inference by Maximising Evidence: Zero-Shot Imitation from Observation with World Models. In Thirty-Seventh Conference on Neural Information Processing Systems, 2023.

---

> > ### Comment · Reviewer_WXaM · 2025-04-14
> >
> > Thank you for the detailed response. The authors have mostly addressed my concerns. My remaining concern is that now it is clearer that the main contribution is positioned to be the error refactorization to EKB and DKB, and less methods that address the errors, the theoretical analysis of the errors is somewhat lacking. Would it be possible to derive some upper bounds on EKB or DKB, conditioned on bounds applied to the policy space? I'd definitely recommend acceptance if this aspect is addressed.

---

> > > ### Author Response · Authors · 2025-04-17
> > >
> > > Thanks for your suggestion. We have added the upper bounds for EKB and DKB in Appendix J. We hope the upper bounds address your concern and give some additional insights about how the proposed method can bridge the two barriers.

---

### Comment · Action_Editor_ZSH7 · 2025-02-24
**3 reviews are in now.**

Dear Authors,

3 reviews are in now.
Specific questions and change requests were provided.
Please carefully read them and send us your rebuttal.

--AE

---

### Author Response · Authors · 2025-02-26
**General response**

Dear AE and Reviewers,

We would like to thank you for your engagement in evaluating our work. We have provided a revised version with all the modifications marked in blue. For each reviewer, we have a separate rebuttal for each of your questions. Please kindly check our responses and new content, and raise any further questions if there are any.

Best regards,
Authors

---

### Decision · Action_Editor_ZSH7 · 2025-04-21

**Recommendation:** Accept as is

**Comment:**

Thank you for the detailed rebuttals for the reviewers' comments and revision of the manuscript. The authors addressed the concerns raised by the reviewers appropriately and all reviewers are satisfied with the revised manuscript. Therefore, I recommend its acceptance as it is. Congratulations!

**Audience:**

Online imitation learning with pretrained models is an important and challenging task.
Due to the popularity of the topic, this submission will attract a broader attention from the TMLR community.

**Claims And Evidence:**

The reviewers found that the problem of imitation learning from observation is important and the proposed framework of decomposing the imitation gap into the embodiment knowledge barrier and demonstration knowledge barrier is novel.
The authors proposed to cope with this challenge by online interactions and surrogate rewards, which is well supported by extensive experiments.
Theoretical justification is left as future work, but the current paper is acceptable due to its novel conceptual contribution and sufficient experimental evaluations.